# MASTER SKILL LEARNING WITH POLICY-GROUNDED SYNERGY OF LLM-BASED REWARD SHAPING AND EXPLORING

**Yanbin Chang[1]\*, Junfan Lin[2]\*, Jie Jiang[1], Runhao Zeng[3], Changxin Huang[1]†, Jianqiang Li[1]**
[1]Shenzhen University, [2]Peng Cheng Laboratory, [3]Shenzhen MSU-BIT University
changyanbin2023@email.szu.edu.cn, linjf@pcl.ac.cn,
2410673027@mails.szu.edu.cn, zengrh@smbu.edu.cn,
huangchx@szu.edu.cn, lijq@szu.edu.cn

## ABSTRACT

The acquisition of robotic skills via reinforcement learning is crucial for advancing embodied intelligence, but designing effective reward functions for complex tasks remains challenging. Recent methods that use large language models to generate reward functions from language instructions can reduce manual effort, but they often produce overly goal oriented rewards that neglect state exploration and cause agents to get stuck in local optima. Traditional RL addresses this by adding exploration bonuses, but these are typically generic and inefficient, wasting resources on exploring task-irrelevant areas. To address these limitations, we propose **Po**licy-grounded Synergy of **R**eward **S**haping and **E**xploration (**PoRSE**), a unified framework that uses large language models to generate task-aware goal rewards and to construct an abstract affordance state space that guides structured and relevant exploration. PoRSE integrates these components through an in-policy improvement process that continuously proposes, evaluates, and filters reward and exploration configurations without requiring policies to train from scratch. This synergy enables more efficient policy learning and stabilizes continual optimization. Experiments across 24 robotic manipulation and locomotion tasks show that PoRSE consistently outperforms prior state-of-the-art LLM-based reward-design methods and achieves the first successes on several previously unsolved challenging tasks.

## 1 INTRODUCTION

Training embodied robots with deep reinforcement learning (DRL) has achieved promising results in acquiring specific skills Celik et al. (2024); Tang et al. (2024), with applications in autonomous driving Coelho et al. (2024), robotic arm manipulation Wen et al. (2025), and legged locomotion control Liang et al. (2024). For example, in legged robot walking tasks, speed rewards encourage forward movement, while stability rewards penalize posture deviations Liang et al. (2024). However, designing effective reward functions typically requires extensive domain knowledge and manual tuning by human experts, making the process time-consuming and labor-intensive Booth et al. (2023).

The recent success of large language models (LLMs) Liang et al. (2023); Minaee et al. (2024); Brohan et al. (2023) and their ability to perform logic reasoning opened new possibilities for reward design in robotic skill learning Zeng et al. (2024); Fu et al. (2024); Venuto et al. (2024). Language to Reward (L2R) Yu et al. (2023) generates modular reward function compositions based on natural language instructions and then uses Model Predictive Control (MPC) to solve robot actions online. Eureka Ma et al. (2023) employs an evolutionary search framework to iteratively optimize reward functions, outperforming human-designed rewards across various robotic tasks. To enhance training efficiency, ROSKA Huang et al. (2025) proposes a reward-policy co-evolution strategy that leverages

---

*Equal Contribution
†Corresponding Author

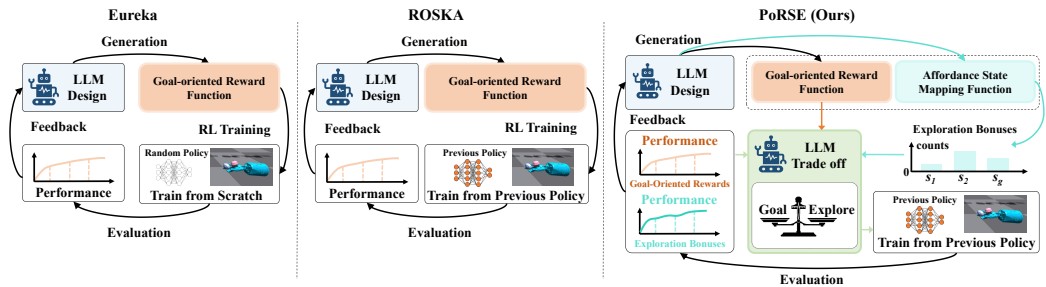

Figure 1: Comparison of PoRSE with prior methods. Eureka Ma et al. (2023) uses LLMs to design reward functions and optimizes them via RL feedback. ROSKA Huang et al. (2025) co-evolves rewards and policies based on historical policies. Our PoRSE guides LLMs to create task-aware rewards and constructs an abstract functional space for efficient exploration.

the knowledge of previously trained policies without the need to start from scratch when rewards are adjusted.

Current LLM-driven approaches, though advanced, rely solely on task-specific textual descriptions for reward design, leading to goal-oriented preferences while neglecting state exploration Chen et al. (2022); Devidze et al. (2022). This limitation is particularly problematic in high-degree-of-freedom robotic tasks, such as dexterous manipulation involving object throwing and catching, where the policy search space is vast and target states are sparse, causing agents to get stuck in local optima. While exploration bonuses can partially address this issue Pathak et al. (2017); Tang et al. (2017), existing methods Ostrovski et al. (2017) do not link exploration to task objectives, resulting in inefficient exploration of irrelevant states Wang et al. (2024). Moreover, conventional RL methods require manual tuning of the trade-off ratio between goal-oriented behavior and state-space exploration for each task, with this ratio fixed from the start of policy training regardless of subsequent policy progress.

To tackle the previously discussed challenges, we introduce **Po**licy-grounded Synergy of **R**eward **S**haping and **E**xploration (**PoRSE**), a novel, unified framework, as shown in Fig. 1. PoRSE empowers LLMs to create task-specific reward functions and simultaneously constructs an abstract affordance state space (AFS) for efficient exploration. In particular, PoRSE goes beyond traditional task-unrelated exploration bonus methods. It harnesses LLMs to design an affordance state space, which condenses high-dimensional environmental states into a low-dimensional, discrete space relevant to the task at hand. By tracking the frequency of visits to these task-relevant AFS, we formulate a curiosity-driven exploration bonus that is tightly aligned with the task objectives as well as the policy behavior.

Given the near-infinite combinatorial space of goal-oriented reward functions and affordance state spaces, exhaustively training policies for each combination to find the best combination as well as its trade-off ratio is infeasible. To address this, PoRSE employs an in-policy-improvement grounding process (IPG), which leverages real-time policy performance feedback to guide LLMs in refining reward-bonus configurations and dynamically balancing their trade-offs. Firstly, IPG incorporates an LLM-bootstrapping elimination-expansion filtering mechanism to efficiently search for optimal reward-bonus combinations and ratio trade-offs across policy improvement stages. Such a filtering mechanism is achieved via coordinate optimization—fixing one variable while optimizing another. By doing so, it simplifies the complex and intertwined search process, greatly reducing the search space and difficulty. Secondly, to effectively integrate the new knowledge acquired during the IPG process into the existing policy, IPG adopts a policy fusion approach similar to that in ROSKA Huang et al. (2025). However, we enhance it by using fast LLM-aided optimization to streamline the determination of the fusion ratio, eliminating the need for cumbersome Bayesian optimization.

Our PoRSE framework masters skill acquisition by integrating goal-oriented reward functions and affordance state space into the IPG process, creating a self-reinforcing cycle where reward shaping, exploration, and policy refinement enhance each other. Experiments on 24 robotic tasks, from

dexterous manipulation to legged locomotion, show PoRSE achieves significant average return improvements and gains breakthroughs in two previously unsolved complex manipulation tasks, setting new milestones in robotic skill acquisition.

## 2 RELATED WORK

**Reward Design for Robot Skill Learning.** Designing effective reward functions remains a core challenge in reinforcement learning (RL) Eschmann (2021). Traditional approaches heavily rely on manual design, requiring extensive domain expertise Booth et al. (2023); Eschmann (2021). Recent studies have explored leveraging large language models (LLMs) to automate reward design Kwon et al. (2023); Zhou et al. (2024); Cao et al. (2024). The Eureka framework Ma et al. (2023) employs LLMs to generate executable reward codes, achieving human-level performance across diverse robotic tasks. ROSKA Huang et al. (2025) enhances data efficiency through reward-policy co-evolution. REvolve Hazra et al. (2024) proposes an evolutionary framework that leverages LLMs and human feedback to iteratively refine reward functions. However, the rewards designed by existing methods lack a certain exploration mechanism. In contrast, our method introduces an efficient exploration reward that can dynamically adjust between rewards as the policy is optimized, achieving efficient training of the policy.

**Curiosity-driven exploration.** In sparse reward tasks, direct policy learning faces challenges of insufficient exploration Ng et al. (1999); Hare (2019); Li et al. (2020). Traditional solutions rely on manually designing dense reward functions, but this process is both time-consuming and brittle. Count-based intrinsic reward mechanisms incentivize exploration by rewarding less-visited states based on visitation frequency statistics Kolter & Ng (2009); Tang et al. (2017). Existing methods set exploration bonuses independently of task objectives, causing agents to waste resources exploring task-irrelevant states. In contrast, PoRSE employs task-relevant state exploration rewards, effectively addressing this limitation by directing exploration toward states that genuinely contribute to task performance.

## 3 PRELIMINARY

**RL-based Robot Skills Acquisition.** Skill learning for robots via reinforcement learning can be formalized as a Markov Decision Process (MDP), where the agent interacts with the environment $E$. The MDP is defined by a quintuple $(S, A, P, R, \gamma)$, consisting of the state space $S$, action space $A$, transition probability $P$, reward $R$, and discount factor $\gamma \in (0, 1]$. At each time step $t$, the robot observes state $s_t \in S$, selects action $a_t \in A$ according to policy $\pi(s_t)$, transitions to $s_{t+1} \sim P(\cdot|s_t, a_t)$, and receives reward $r_t = R(s_t, a_t, s_{t+1})$. The return for state $s_t$ is the cumulative $\gamma$-discounted reward: $\sum_{i=t}^{T} \gamma^{i-t} r_i$. The objective of policy optimization is to maximize the expected cumulative return. We define $\theta_p$ as the policy parameters after $p$ updates: $\theta_p = \mathcal{I}(R, \theta_0, p)$, where $\theta_0$ are the randomly initialized policy parameters, and $\mathcal{I}(R, \theta, p)$ represents the policy improvement process given reward function $R$ over $p$ environment steps.

**Reward Function Generation via LLMs.** Large Language Models (LLMs) have shown strong capabilities in code generation and reasoning for robotic learning tasks, enabling automated reward function generation in reinforcement learning. The Eureka framework Ma et al. (2023) uses LLMs to design reward functions $R_g$ based on task goal descriptions $I_d$ and environment code $I_e$. Eureka employs a multi-iteration optimization approach: in the $n$-th iteration, the best reward function $R_{g,\text{best}}^{n-1}$ from the previous iteration and the sparse reward evaluation $V(\pi)$ of the trained policy $\pi$ are fed back to the LLM to generate optimized reward functions, as shown in Eq. equation 1:

$$\mathbf{R}_g^n = \mathcal{LLM}(I_d, I_e, R_{g,\text{best}}^{n-1}, V(\pi)), \tag{1}$$

where $V(\pi)$ is a sparse reward array reflecting the policy's optimization trend. Each generated reward function trains a policy from scratch for $T_{\max}$ epochs, and the reward function yielding the highest task performance is selected as the best for this iteration, as shown in Eq. equation 2:

$$R_{g,\text{best}}^n = \underset{R_{g,k}^n \in \mathbf{R}_g^n}{\arg\max} V\left(\mathcal{I}(R_{g,k}^n, \theta_0, T_{\max})\right). \tag{2}$$

Through this iterative process, the LLM improves reward functions based on feedback, enhancing robotic skill learning performance.

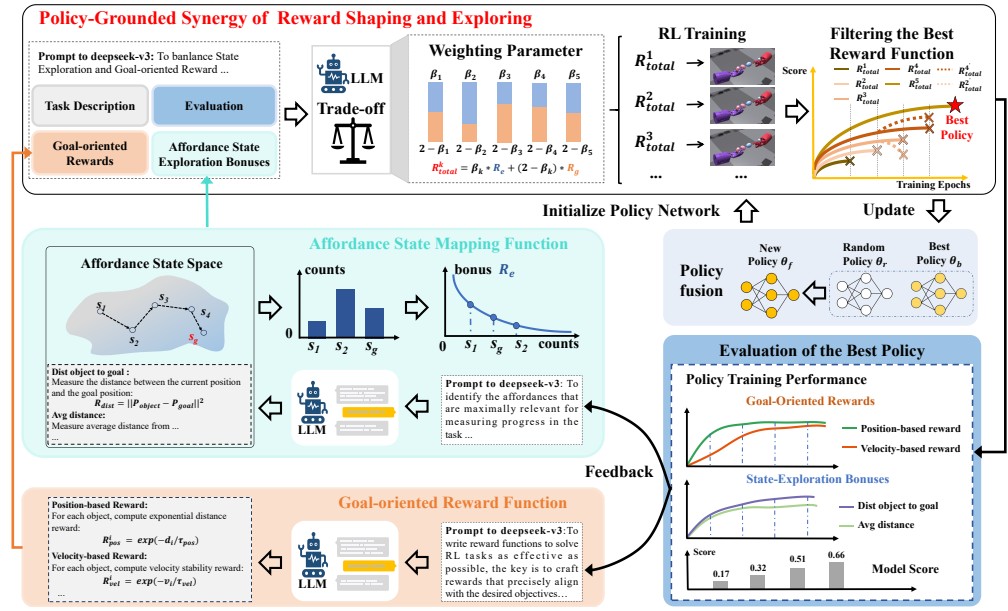

Figure 2: Overview of PoRSE. It leverages LLMs to generate goal-oriented rewards while building an affordance mapping function for exploration bonuses. These rewards are dynamically combined to optimize policies. An iterative feedback loop continuously refines rewards and affordance state space, creating a co-evolutionary system.

# 4 METHOD

The methodology section is organized as follows: Sec . 4.1 introduces an abstract affordance state space for an efficient state-exploration bonus, which is combined with goal-oriented rewards for policy training. Sec . 4.2 proposes a dynamic exploration-exploitation adjustment mechanism guided by policy feedback. To avoid costly training of numerous reward-bonus combinations, we employ policy inheritance and a tournament elimination mechanism. Sec . 4.3 deepens the interplay between goal-oriented behavior, state exploration, and policy learning, showing how their collaborative optimization accelerates skill acquisition. The overall framework of the method is shown in Fig. 2. The following sections detail each component.

## 4.1 GOAL-RELEVANT EXPLORATION BONUS GENERATION

### 4.1.1 AFFORDANCE STATE VISITING COUNT AS EXPLORATION BONUS.

To prevent the robot from exploring irrelevant states and ensure efficient policy optimization, we introduce a task-relevant exploration bonus based on a curiosity-driven method Pathak et al. (2017); Tang et al. (2017). As shown in Fig. 2, we map the robot's state space $S$ to a low-dimensional *Affordance State Space* (AFS) $S_o$ using a mapping function $\mathcal{M} : S \rightarrow S_o$. Each dimension of $S_o$ quantifies the distance between the robot's current state and the goal state based on specific behavioral affordances. For example, in the *DoorOpenInward* task, $S_o = [s_o^d, s_a^l]$, where $s_o^d$ is the Euclidean distance to the door handle, and $s_a^l$ is the door's angular displacement. We design the exploration bonus by computing the agent's visitation frequency to each state in the AFS. Frequently visited states receive lower bonuses, while less-explored states receive higher bonuses. To facilitate counting, we discretize the continuous AFS using a function $\mathcal{D}(S_o)$, resulting in $C$ discrete states $S_{o,d} = \mathcal{D}(S_o)$. Accordingly, our exploration bonus $R_e$ is defined as follows:

$$R_{\mathrm{e}}(s_o) = \frac{\lambda}{\sqrt{\sum_{t=1}^{T} \mathbb{I}(s_{o,d}^t = s_{o,d}^c)}}. \tag{3}$$

Here, $\sum_{t=1}^{T} \mathbb{I}(s_{o,d}^t = s_{o,d}^c)$ counts the number of times the state $s_{o,d}^c$ has been visited, and $\lambda$ is the weight parameter that regulates the exploration bonus.

However, defining the affordance state space for each robotic task is undoubtedly time-consuming and labor-intensive. To address this, we leverage the task-parsing capabilities of LLMs, using task descriptions as instructions to guide the model in automatically setting the state mapping function $\mathcal{M}$ for each robotic task. This can be defined as: $\mathbf{M}^n = \mathcal{LLM}(I_d, I_e)$.

Given that the exploration bonus is inherently grounded in the affordance state space, which is closely associated with the task, this effectively encourages the agent to explore task-relevant states while mitigating excessive exploration of goal-irrelevant states. We then combine the exploration bonus with the goal-oriented reward $R_g$ from Eq equation 1 , the total reward function is formulated as follows:

$$R_{\text{total}}^k(s) = R_e^k + R_g^k, \quad R_g \in \mathbf{R}_g^n. \tag{4}$$

### 4.1.2 REWARD-BONUS REFINEMENT.

To enhance the quality of goal-oriented reward and state-exploration bonus, following the previous method Eureka Ma et al. (2023), we feed the evaluation data of the policy back to LLMs and perform multi-iteration refinement of the reward and bonus. Specifically, we adopt the iterative optimization approach shown in Equ. (1) to optimize the goal-oriented reward function $R_g$. For the exploration bonus, we similarly guide LLMs to adjust the mapping function $\mathbf{M}^n$ of the AFS based on policy feedback. The key difference lies in the LLM prompt design: instead of using the best reward function from the previous iteration as instructions, we now use the best AFS mapping function $\mathcal{M}_{\text{best}}^{n-1}$ from the previous iteration as a reference sample. Formally:

$$\mathbf{M}^n = \mathcal{LLM}(I_d, I_e, \mathcal{M}_{\text{best}}^{n-1}, V(\theta)), \tag{5}$$

This iterative process enables co-evolution between the reward functions and mapping functions, progressively improving the exploration efficiency of the RL agent.

### 4.2 IN-POLICY-IMPROVEMENT GROUNDING PROCESS (IPG)

To balance the state-exploration bonuses $R_e$ and goal-oriented rewards $R_g$ for policy improvement, we employ a weighted summation approach, formulating the total reward as follows:

$$R_{\text{total}}^k = \beta * R_e^k + (2 - \beta)R_g^k, \quad \beta \in [0, 2], R_g \in \mathbf{R}^n. \tag{6}$$

where $\beta$ serves as the weighting parameter to balance the contributions of the two components. Consequently, the weighting parameter $\beta$ plays a critical role in balancing the agent's exploration and exploitation behaviors. To determine an proper value for $\beta$, we leverage the data analysis capabilities of LLMs to evaluate the policy performance feedback $V(\theta)$, thereby generating a set of potential $\beta$ values: $\mathbf{B}^n = \mathcal{LLM}(I_d, V(\theta))$. $\mathbf{B}^n$ denotes a set comprising $K$ distinct $\beta$ values, formally defined as: $\mathbf{B}^n = [\beta_1^n, \beta_2^n, \ldots, \beta_K^n]$.

### 4.2.1 LLM-BOOTSTRAPPING ELIMINATION-EXPANSION FILTERING (LEF).

Since each $\beta$ consumes resources during the continuous optimization of the policy, we introduce an elimination mechanism to reduce the computational resources required for policy optimization, as shown in the top of Fig. 2. We define an exploration-goal reward pair as a combination of a state-exploration bonus $R_e$, a goal-oriented reward $R_g$, and a reward weight coefficient $\beta$. A set of such pairs is denoted as $\mathbf{N} = \{(\beta_1, R_{e,1}, R_{g,1}), \cdots, (\beta_N, R_{e,N}, R_{g,N})\}$. During the parallel training of $N$ exploration-goal reward pairs, we conduct an elimination process after every $H$ epochs of all policy training, removing the lowest-performing pair based on the current policy's performance. This process is similar to the linear population reduction mechanism in the L-SHADE algorithm Tanabe & Fukunaga (2014). To further improve the quality of $\beta$, we incorporate an expansion mechanism inspired by the PSO algorithm Priyadarshi & Kumar (2025). Specifically, after the elimination process, the best-performing reward-bonus combination among the survivors is used as the base for expansion, introducing $J$ new mutated $\beta$ values. This cycle of elimination and expansion alternates iteratively, enabling continuous exploration for better $\beta$ configurations.

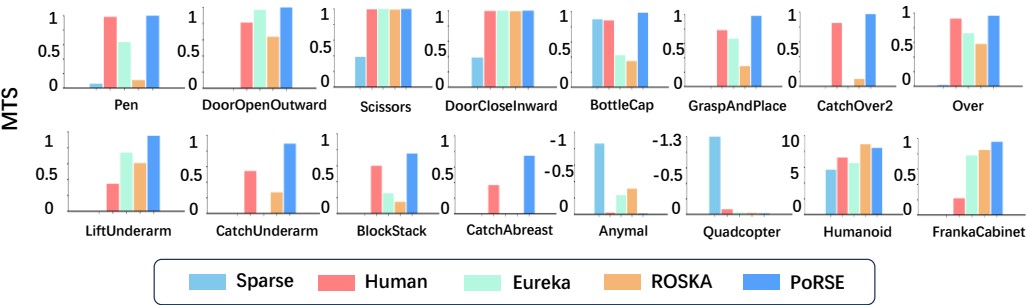

Figure 3: MTS Performance Comparison on Moderate-Difficulty Manipulation Tasks. Compared to other baselines, PoRSE achieved the best results on 15 tasks.

### 4.2.2 LEVERAGE PREVIOUS KNOWLEDGE VIA POLICIES INHERITANCE.

To address the inefficiency of retraining policies from scratch for each reward-bonus refinement iteration, we adopt a strategy similar to ROSKA Huang et al. (2025), blending the best previous policy parameters $\theta_{\text{best}}$ with a stochastic policy $\theta_{\text{random}}$:

$$\theta_f(\alpha) = \alpha \cdot \theta_{\text{best}} + (1 - \alpha) \cdot \theta_{\text{random}}, \tag{7}$$

where $\alpha \in [0, 1]$ is the fusion ratio. ROSKA uses Bayesian optimization to find the optimal $\alpha$, which is computationally expensive due to repeated policy training and data collection.

Instead, we dynamically adjust $\alpha$ using the LEF method: $\alpha_{\text{new}} = \mathcal{LLM}(I_d, V(\theta))$, where $\alpha$ is optimized via elimination and mutation strategies. This approach avoids the need for collecting ratio-performance sample pairs, reducing computational cost while maintaining an effective exploration-exploitation balance.

### 4.3 POLICY-GROUNDED SYNERGY OF REWARD SHAPING AND EXPLORATION

In the PoRSE framework, goal-oriented rewards and state-exploration bonus, and policies are synergistically optimized to facilitate efficient robot skill acquisition. The goal-oriented reward ($R_g$) provides precise guidance for achieving task objectives through reward signals. The state-exploration bonus ($R_e$) is designed within an affordance state space, encouraging the policy to explore unvisited state regions. These two components are fused using a weighting coefficient $\beta$, and dynamically adjust the weighting coefficient $\beta$ based on the real-time performance of the policy. When the policy's performance stagnates or deteriorates, $\beta$ increases to emphasize the exploration bonus, helping the policy escape local optima. Conversely, when the policy demonstrates effective optimization and progressively achieves the task objectives, $\beta$ decreases to strengthen the goal-oriented reward, accelerating the attainment of the target state. Through iterative cycles of synergistic optimization among these three components, the PoRSE framework enables efficient learning of complex skills.

## 5 EXPERIMENT

### 5.1 EXPERIMENTAL SETTINGS

**Training Setting.** In all experiments, we used the PPO algorithm Schulman et al. (2017); Makovi-ichuk & Makoviychuk (2021) for policy training with the DeepSeek-V3 model Liu et al. (2024). Our approach runs $N = 5$ iterative rounds, each generating $K = 6$ reward-mapping pairs for training. In each round, after every $H = 500$ epochs, the lowest-performing policy is eliminated. At the 1500-epoch mark, $J = 5$ mutated pairs are introduced, and the lowest-performing 6 pairs are eliminated. Subsequently, every $H = 500$ epochs, the worst pair is removed. Only one policy completes the full 3000 epochs per round. We alternate between reward coefficient $\beta$-based and policy inheritance coefficient $\alpha$-based mutation searches across iterations. For comparison, Eureka and ROSKA also generate 6 reward functions per round over 5 rounds, with each trained for 3000 epochs. Sparse rewards and human-designed rewards follow the same setup. We record the best

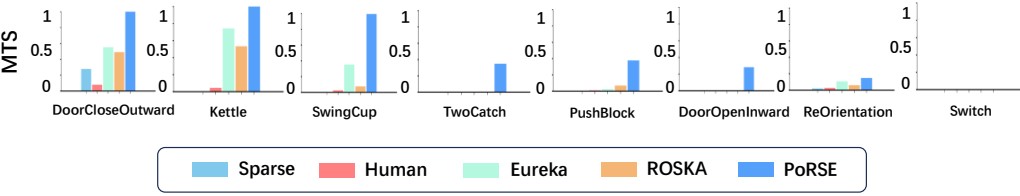

Figure 4: MTS Performance Comparison on Hard Manipulation Tasks. PoRSE achieved the best results on all 8 difficult tasks.

result per experiment using 5 random seeds. The computational cost for our method, Eureka, and ROSKA is nearly identical across experiments. Additional settings are detailed in the appendix.

**Benchmark.** We conducted experimental evaluations on 24 robotic skill learning tasks across two simulation environments: Bi-DexHands Chen et al. (2022) and Isaac Gym Makoviychuk et al. (2021). Specifically, 20 tasks originated from Bi-DexHands, while the remaining 4 were from Isaac Gym. Bi-DexHands serves as a benchmark environment for dexterous bimanual manipulation tasks, focusing on evaluating robotic systems' capability to perform complex manipulation using both hands. In contrast, Isaac Gym provides a benchmark for continuous control in robotics, encompassing skill learning tasks involving legged robots and manipulator arms, among other robotic platforms. In order to improve readability, we have adopted abbreviations for some tasks, and the complete task names are provided in the appendix.

**Evaluation Metrics.** In our experiments, we utilized Maximum Training Success (**MTS**) as the principal evaluation metric. MTS quantifies the highest sparse reward attained during training, serving as a critical indicator of policy efficacy.

**Baseline Methods.** We compared the performance of our method against four baseline approaches: Sparse rewards Ma et al. (2023), human-designed rewards Makoviychuk et al. (2021), Eureka Ma et al. (2023), and ROSKA Huang et al. (2025). **Sparse Rewards** refers to reward settings that express the task objective,and are defined in EurekaMa et al. (2023). **Human Expert-Designed Rewards** are carefully designed by human experts, and more refined compared to sparse rewards. **Eureka** is a general framework that automatically generates reward functions leveraging LLM Ma et al. (2023). **ROSKA** is a robot skill learning framework Huang et al. (2025). **PoRSE** is our method.

### 5.2 COMPARISON RESULTS.

We categorized the 24 reinforcement learning tasks in this study into two difficulty levels—moderate and hard—based on training scores using human-designed rewards. Results for each level are presented in separate tables.

**Performance Comparison on Moderate-Difficulty Manipulation Tasks.** Experimental results in Fig. 3 demonstrate PoRSE's superiority in Maximum Training Success (MTS). PoRSE achieved perfect MTS (1.000) in 5/16 tasks (e.g., Pen, Scissors), outperforming the human baseline by 1.6%-3.5%. In GraspAndPlace, PoRSE reached MTS 0.984 (25.3% better than human baseline 0.785). The only exception was Humanoid locomotion, where Roska (8.917) slightly outperformed PoRSE (8.454, +5.2%), due to bipedal tasks favoring goal-oriented rewards. However, PoRSE excelled in Anymal (MTS -0.012, +42% vs. human baseline -0.021). Overall, PoRSE achieved leading performance in 15/16 tasks, confirming its effectiveness across diverse robotic scenarios.

**Performance on Hard Manipulation Tasks.** In challenging manipulation tasks, PoRSE demonstrated superior performance through its integrated optimization mechanism, as shown in Fig. 4. It achieved perfect MTS (1.000) in DoorCloseOutward and Kettle tasks, outperforming Eureka (0.553, 0.742). As shown in Fig. 4, PoRSE attained a significant improvement over the human baseline in DoorOpenInward, and was the first to solve TwoCatch (MTS 0.349). In BlockStack, PoRSE achieved MTS 0.753, significantly surpassing ROSKA (0.148). These results validate PoRSE's effectiveness in sparse reward scenarios and high-dimensional action spaces.

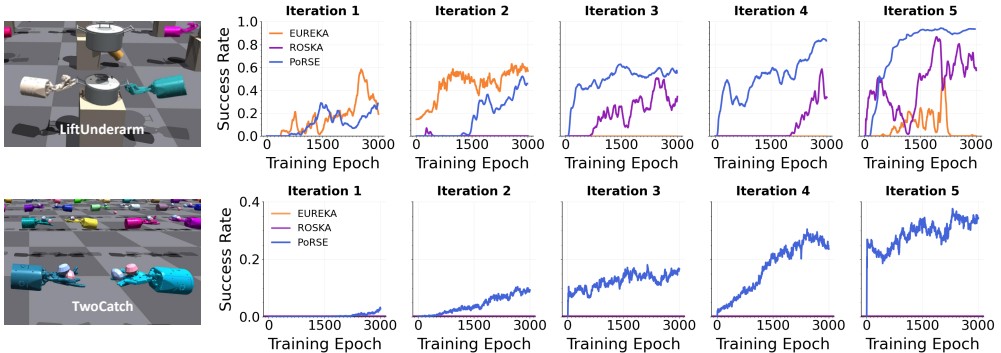

Figure 5: Results show that all methods can guide the policy to optimize in simple tasks, but PoRSE achieved the best results. For complex tasks, only PoRSE can effectively guide policy optimization, and both Eureka and ROSKA methods are ineffective.

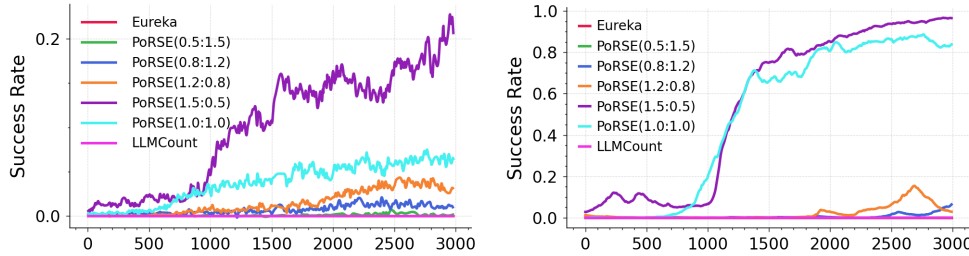

Figure 6: Training curves of different $\beta$ values. It is evident that only by employing appropriate ratios can the policy be successfully trained to attain better performance.

**Iterative Training Performance Comparison.** As shown in Fig. 5, we compare PoRSE's with Eureka and ROSKA on LiftUnderarm (simple) and TwoCatch (complex) tasks. In LiftUnderarm, PoRSE reached a 0.97 success rate by Iteration 5 (1500 episodes), outperforming Eureka (0.6) and ROSKA (0.8) and converging faster. In TwoCatch, where Eureka and ROSKA failed (success=0), PoRSE achieved a 0.4 success rate by Iteration 5, with a tenfold increase from Iteration 1, highlighting its effectiveness in handling sparse rewards and refining policies.

## 5.3 ABLATION STUDIES

**Ablation Study Questions.** This section conducts ablation experiments to investigate: (1) How goal-oriented rewards, state-exploration bonuses, and policy inheritance in PoRSE, and their individual contributions? (2) The roles of reward-bonus coefficient $\beta$ and policy inheritance coefficient $\alpha$, and whether their alternating iteration is necessary? (3) How do different $\beta$ values affect policy optimization, and the relative importance of reward components? (4) Is PoRSE's iterative reward optimization process necessary? (5) How robust is PoRSE when the Affordance State Space is noisy or randomly assembled?

**Component-wise Ablation Analysis.** This ablation study examines the roles of goal-oriented rewards ($R_g$), exploration-oriented bonuses ($R_e$), and dynamic policy fusion inheritance ($\theta_{fusion}$) in PoRSE. As shown in Tab. 1, removing $R_g$ significantly degrades performance in precision tasks like TwoCatch ($0.349 \rightarrow 0.190$) while retaining reasonable performance in manipulation tasks like FrankaCabinet (0.883, -7.7%). Removing $R_e$ reduces performance in exploration-requiring tasks, with BlockStack dropping from 0.753 to 0.393 and TwoCatch from 0.349 to 0.193. Eliminating $\theta_{fusion}$ causes severe performance drops in long-term optimization tasks, with FrankaCabinet decreasing from 0.957 to 0.671 and TwoCatch plummeting from 0.349 to 0.017. The synergistic effects of all components are critical, as removing any single component reduces PushBlock success

Table 1: The MTS results show that both the architectural components and optimization strategy contribute to the final performance.

| Method | Anymal | | Quadcopter | | Franka | |
|---|---|---|---|---|---|---|
| | MTS | Gap | MTS | Gap | MTS | Gap |
| PoRSE w/o $R_g$ | $-0.128$ | $\downarrow 0.116$ | $-0.040$ | $\downarrow 0.026$ | 0.883 | $\downarrow 0.074$ |
| PoRSE w/o $R_e$ | $-0.097$ | $\downarrow 0.085$ | $-0.038$ | $\downarrow 0.024$ | 0.912 | $\downarrow 0.045$ |
| PoRSE w/o $\theta_{fusion}$ | $-0.346$ | $\downarrow 0.334$ | $-0.034$ | $\downarrow 0.020$ | 0.671 | $\downarrow 0.286$ |
| PoRSE w/o $R_{ratio}$ | $-0.066$ | $\downarrow 0.054$ | $-0.019$ | $\downarrow 0.005$ | 0.946 | $\downarrow 0.011$ |
| PoRSE w/o $\theta_{ratio}$ | $-0.020$ | $\downarrow 0.008$ | $-0.015$ | $\downarrow 0.001$ | 0.940 | $\downarrow 0.017$ |
| PoRSE | $-0.012$ | | $-0.014$ | | 0.957 | |

Table 2: The MTS results show that both the architectural components and optimization strategy contribute to the final performance.

| Method | BlockStack | | PushBlock | | TwoCatch | |
|---|---|---|---|---|---|---|
| | MTS | Gap | MTS | Gap | MTS | Gap |
| PoRSE w/o $R_g$ | 0.328 | $\downarrow 0.425$ | 0.295 | $\downarrow 0.023$ | 0.190 | $\downarrow 0.159$ |
| PoRSE w/o $R_e$ | 0.393 | $\downarrow 0.360$ | 0.306 | $\downarrow 0.012$ | 0.193 | $\downarrow 0.156$ |
| PoRSE w/o $\theta_{fusion}$ | 0.603 | $\downarrow 0.150$ | 0.236 | $\downarrow 0.082$ | 0.017 | $\downarrow 0.332$ |
| PoRSE w/o $R_{ratio}$ | 0.296 | $\downarrow 0.457$ | 0.243 | $\downarrow 0.075$ | 0.297 | $\downarrow 0.052$ |
| PoRSE w/o $\theta_{ratio}$ | 0.590 | $\downarrow 0.163$ | 0.309 | $\downarrow 0.009$ | 0.276 | $\downarrow 0.073$ |
| PoRSE | 0.753 | | 0.318 | | 0.349 | |

Table 3: MTS comparative results of PoRSE and LLMCount baselines across six representative manipulation tasks. PoRSE achieves apparent improvement in task success rates.

| Method | Pen | TwoCatch | Franka | BlockStack | PushBlock | DoorOpenInward |
|---|---|---|---|---|---|---|
| LLMCount | 0.412 | 0.000 | 0.706 | 0.140 | 0.127 | 0.025 |
| PoRSE | **1.000** | **0.349** | **0.957** | **0.753** | **0.318** | **0.250** |

rates by 7.2%-25.8%, while the complete PoRSE framework achieves optimal performance through collaborative optimization.

**Optimization Strategy Ablation Study.** This study investigates the effects of alternating reward fusion ($\beta$) and policy inheritance ($\alpha$) coefficients during iterative optimization. Fixing either coefficient degrades MTS across tasks. As shown in Tab. 1 and Tab. 2, with fixed policy fusion inheritance (PoRSE w/o $\theta_{fusion}$), the "TwoCatch" MTS drops from 0.349 to 0.276 due to reduced adaptability. Fixing reward fusion ($R_{ratio}$) disrupts exploration-exploitation balance, as seen in the "PushBlock" task where MTS falls from 0.318 to 0.243. The complete PoRSE method, alternating both coefficients, achieves optimal reward-policy coordination, delivering the best performance across all tasks.

**Ablation Analysis of $\beta$.** We evaluated different reward ratios ($R_g : R_e$) for DoorOpenInward and TwoCatch tasks. Configurations included Eureka ($R_g$-only), LLMCountSarukkai et al. (2024) ($R_e$-only), and PoRSE ratios (0.5:1.5, 0.8:1.2, 1.2:0.8, 1.5:0.5, 1.0:1.0), all trained for 3000 epochs. As shown in Fig. 6, results show single-reward approaches failed completely (0% success), while optimal fusion ratios achieved significant improvements. PoRSE (1.5:0.5) reached 97% success on DoorOpenInward and 25% on TwoCatch, outperforming prior methods ($\sim 0\%$).

Table 4: MTS Performance Comparison of PoRSE-AFS-Random and Baseline Methods on Representative Manipulation Tasks (mean ± standard deviation).

| Task | BlockStack | PushBlock | LiftUnderarm |
|---|---|---|---|
| Sparse | 0.000 ± 0.001 | 0.003 ± 0.004 | 0.000 ± 0.000 |
| Human | 0.600 ± 0.229 | 0.011 ± 0.003 | 0.348 ± 0.140 |
| Eureka | 0.254 ± 0.119 | 0.025 ± 0.008 | 0.739 ± 0.166 |
| Roska | 0.148 ± 0.154 | 0.069 ± 0.049 | 0.608 ± 0.211 |
| PoRSE-AFS-Random | 0.680 ± 0.080 | 0.324 ± 0.034 | 0.802 ± 0.029 |
| PoRSE | **0.753 ± 0.210** | **0.378 ± 0.022** | **0.952 ± 0.015** |

**Ablation Study on Reward Refinement.**

This study examines the impact of reward and bonus refinement during policy optimization. Using LLMCount Sarukkai et al. (2024) as a baseline, which generates a fixed exploration bonus, we compare its performance against PoRSE. As shown in Tab. 3, results show PoRSE significantly outperforms LLMCount across all tasks, demonstrating that static rewards inadequately address complex skill learning needs. For example, in the TwoCatch task, LLMCount's fixed bonuses failed to adapt to policy performance changes, while PoRSE's collaborative optimization loop dynamically adjusted $R_e$ and $R_g$ ratios, effectively guiding training through different learning stages and highlighting the benefits of dynamic reward refinement.

**Ablation Study on AFS Robustness.** This study evaluates the impact of replacing the LLM-generated task-aware AFS with a randomly assembled AFS. As shown in Tab. 10, the variant PoRSE-AFS-Random remains clearly stronger than Sparse, Human, Eureka, and ROSKA on the representative manipulation tasks BlockStack, PushBlock, and LiftUnderarm. For example, it achieves $0.680 \pm 0.080$ on BlockStack versus $0.254 \pm 0.119$ for Eureka and $0.148 \pm 0.154$ for ROSKA, and reaches $0.324 \pm 0.034$ on PushBlock where the best baseline is $0.069 \pm 0.049$. Although it is below the full PoRSE that uses task-relevant AFS, the elimination mechanism prunes unhelpful dimensions and the LLM adapts the exploration coefficient, which preserves strong performance despite noisy AFS and confirms that PoRSE is robust to imperfect affordance specifications.

## 6    CONCLUSION

We propose Policy-grounded Synergy of Reward Shaping and Exploration (PoRSE), a novel framework for balancing reward design and state exploration in robotic skill learning. PoRSE integrates LLM-generated affordance state spaces with a dynamic curiosity-driven bonus mechanism, enabling efficient exploration aligned with task objectives without manual reward tuning. The in-policy-improvement grounding process (IPG) optimizes reward configurations via real-time policy feedback, using fast LLM-aided methods instead of manual Bayesian optimization. Experiments on diverse robotic tasks show PoRSE's superiority, achieving significant performance gains over state-of-the-art methods. By unifying reward shaping, exploration, and policy refinement in a self-reinforcing loop, PoRSE sets a new paradigm for autonomous robotic skill acquisition.

## 7    LIMITATION

Similar to other LLM-based generation approaches, our method is also subject to the inherent instability of LLM outputs. Even with identical prompts, the quality of generated code can vary, and non-executable code may occasionally be produced. This is a common limitation of LLMs rather than a specific issue of our approach. Nevertheless, with the rapid advancement and continuous iteration of LLM technology, stability in reward design and code generation is expected to gradually improve, and LLM capabilities in these areas will continue to strengthen. To ensure a fair comparison in this paper, we have implemented the following measure for Eureka, ROSKA, LLMCount, and PoRSE: all six functions generated in each iteration undergo code correctness verification. This ensures that every generated function is executable, thereby mitigating result deviations caused by LLM instability and maintaining the reliability and fairness of the comparative evaluation.

# 8 ACKNOWLEDGE

This work is supported in part by the National Natural Science Foundation of China (No. 62403325, No. 62325307, No. 62527809, No. 62203134, No. 62373258, No. 62506180), in part by the Natural Science Foundation of Guangdong Province (No. 2023B1515120038), in part by Shenzhen Science and Technology Innovation Commission (No. 20231122104038002, No. KJZD20230923113801004, No. JCYJ20240813141628038, No. KJZD20230923115215032), in part by the Shenzhen Key Industry R&D Program Project (No. ZDCY20250901102300001), in part by China Postdoctoral Science Foundation (No. 2025M771522), in part by the Major Key Project of PCL (No. PCL2024A04, No. PCL2025A17).

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

# A  APPENDIX

## A.1  ALGORITHM DESCRIPTION

In this section, we present the Policy-grounded Synergy of Reward Shaping and Exploration (PoRSE) framework, as detailed in Algorithm 1.

The algorithm operates through $N = 5$ iterations of policy optimization. In each iteration, the large language model (LLM) generates $K = 6$ combinations containing goal-oriented reward functions $\mathbf{R}^n$ and state-exploration affordance mapping functions $\mathbf{M}^n$, where $n$ denotes the iteration index. The exploration bonus is calculated using a parameter $\lambda$=0.01 to scale the reward derived from state visitation counts. Each candidate combination trains parallel policies via Proximal Policy Optimization (PPO) for up to $T_{\max} = 3000$ epochs, using dynamically fused rewards $R_{\text{total}}^k = \beta^k R_{\text{goal}}^k + (2-\beta^k) R_{\text{explore}}^k$. Here, the reward coefficient $\beta$ (default: 1) balances goal-oriented rewards and state-exploration bonuses, while the policy inheritance ratio $\alpha$ (default: 0.8) controls fusion with prior policies. During training, an LLM-boostrapping elimination-expansion mechanism prunes candidates: the worst-performing policies are removed every 500 epochs, while at the 1500-epoch mid-stage, PoRSE generates $J = 5$ mutated variants of either $\beta$ (odd iterations) or $\alpha$ (even iterations) to expand the candidate pool. After mutation, the bottom 6 policies are eliminated, and pruning continues every 500 epochs until only the best policy $\theta_{\text{best}}^n$ remains at 3000 epochs.

---

**Algorithm 1** Policy-grounded Synergy of Reward Shaping and Exploration (PoRSE)

---

**Require:** Task description $I_d$, environment code $I_e$, iterations $N = 5$, candidates per iteration $K = 6$, max epochs $T_{\max} = 3000$, initial policy $\theta_0$

1: **for** $n = 1$ **to** $N$ **do**
2:   **if** $n = 1$ **then**
3:     // Reward and mapping functions initialization via LLM
4:       $\mathbf{R}^n \leftarrow \mathcal{LLM}(I_d, I_e), \quad \mathbf{M}^n \leftarrow \mathcal{LLM}(I_d, I_e)$
5:   **else**
6:     // Reward and mapping functions refinement via LLM
7:       $\mathbf{R}_g^{n+1} \leftarrow \mathcal{LLM}(I_d, \mathcal{R}_{\text{best}}^n, V(\theta_{\text{best}}^n)), \quad \mathbf{M}^{n+1} \leftarrow \mathcal{LLM}(I_d, \mathcal{M}_{\text{best}}^n, V(\theta_{\text{best}}^n))$
8:   **end if**
9:   // Construct goal-oriented rewards and state-exploration bonuses
10:   **for** $k = 1$ **to** $K$ **do**
11:     $R_{g,k}^n \leftarrow R_k^n(s, a), \quad R_{e,k}^n \leftarrow \lambda / \sqrt{\text{count}(\mathcal{D}(\mathcal{M}_k^n(s)))}$
12:   **end for**
13:   // Parallel policy training using fused rewards
14:   **for** $(R_{g,k}^n, R_{e,k}^n) \in \mathbf{N}^n$ **do**
15:     $R_{\text{total}}^k \leftarrow \beta R_{e,k}^n + (2 - \beta) R_{g,k}^n, \quad \theta_f^k \leftarrow \alpha \theta_{\text{best}}^{n-1} + (1 - \alpha)\theta_0$ {Default $\alpha$ and $\beta$}
16:     Train $\theta_f^k$ with PPO
17:   **end for**
18:   // LLM-bootstrapping Elimination-expansion Filtering
19:   **for** t=500,1000,…,$T_{max}$ **do**
20:     **if** $t = 1500$ **then**
21:       Generate $J = 5$ mutated $\alpha$ or $\beta$ via PSO,    Remove the worst 6 policy $\theta_f$
22:     **else**
23:       Remove the worst policy $\theta_f$
24:     **end if**
25:   **end for**
26: **end for**
27: **return** $\theta_{\text{best}}$

---

In later iterations ($n \geq 2$), the LLM refines new reward functions $\mathbf{R}^{n+1}$ and affordance mappings $\mathbf{M}^{n+1}$ based on sparse reward evaluations $V(\theta_{\text{best}}^n)$ from the current best policy. This cycle repeats for all 5 iterations, with $\beta$ and $\alpha$ alternately mutated to diversify exploration. The final output is an optimized policy $\theta_{\text{best}}$ that synergistically balances goal achievement and state-space exploration.

## A.2 EXTENDED EXPERIMENT RESULTS

### A.2.1 DETAILED EXPERIMENT RESULTS

As shown in Tab. 5 and Tab. 6, we present the detailed results of all experimental data involved in this paper in the form of mean ± standard deviation, with each result derived from experiments with 5 different random seeds.

Table 5: MTS performance comparison on moderate-difficulty tasks. MTS (mean ± std) showing PoRSE's superiority over baselines in manipulation and locomotion tasks.

| Task name | Sparse | Human | Eureka | Roska | PoRSE |
|---|---|---|---|---|---|
| **Pen** | 0.061±0.108 | 0.984±0.016 | 0.634±0.460 | 0.111±0.077 | **1.000±0.000** |
| **DoorOpenOutward** | 0.000±0.000 | 0.813±0.418 | 0.971±0.061 | 0.638±0.303 | **1.000±0.000** |
| **Scissors** | 0.387±0.529 | 0.996±0.001 | **1.000±0.000** | 0.992±0.011 | **1.000±0.000** |
| **DoorCloseInward** | 0.400±0.547 | **1.000±0.000** | **1.000±0.000** | **1.000±0.000** | 1.000±0.000 |
| **BottleCap** | 0.901±0.033 | 0.886±0.056 | 0.424±0.145 | 0.348±0.175 | **0.988±0.011** |
| **GraspAndPlace** | 0.001±0.001 | 0.785±0.298 | 0.667±0.079 | 0.284±0.084 | **0.984±0.008** |
| **CatchOver2** | 0.000±0.000 | 0.854±0.013 | 0.001±0.001 | 0.100±0.221 | **0.973±0.008** |
| **Over** | 0.019±0.042 | 0.925±0.011 | 0.725±0.377 | 0.579±0.388 | **0.965±0.013** |
| **LiftUnderarm** | 0.000±0.000 | 0.348±0.140 | 0.739±0.166 | 0.608±0.211 | **0.952±0.015** |
| **CatchUnderarm** | 0.000±0.000 | 0.544±0.250 | 0.000±0.000 | 0.271±0.370 | **0.894±0.038** |
| **BlockStack** | 0.000±0.001 | 0.600±0.229 | 0.254±0.119 | 0.148±0.154 | **0.753±0.210** |
| **CatchAbreast** | 0.000±0.000 | 0.369±0.213 | 0.000±0.000 | 0.000±0.000 | **0.745±0.073** |
| **FrankaCabinet** | 0.000±0.010 | 0.100±0.050 | 0.778±0.175 | 0.850±0.329 | **0.957±0.049** |
| **Anymal** | -0.863±0.017 | -0.021±0.005 | -0.235±0.306 | -0.314±0.540 | **-0.012±0.005** |
| **Quadcopter** | -1.382±0.083 | -0.067±0.029 | -0.023±0.007 | -0.019±0.007 | **-0.014±0.005** |
| **Humanoid** | 5.691±0.880 | 7.235±1.152 | 6.534±1.933 | **8.917±0.512** | 8.454±0.343 |

Table 6: MTS performance comparison on hard manipulation tasks. Results demonstrate PoRSE's breakthroughs in hard skill learning tasks where existing methods fail completely.

| Task name | Sparse | Human | Eureka | Roska | PoRSE |
|---|---|---|---|---|---|
| **DoorCloseOutward** | 0.279±0.348 | 0.244±0.424 | 0.553±0.340 | 0.491±0.467 | **1.000±0.000** |
| **Kettle** | 0.001±0.002 | 0.046±0.077 | 0.742±0.419 | 0.534±0.448 | **1.000±0.000** |
| **SwingCup** | 0.001±0.003 | 0.025±0.026 | 0.353±0.407 | 0.076±0.043 | **0.995±0.002** |
| **TwoCatchUnderarm** | 0.000±0.000 | 0.000±0.000 | 0.001±0.001 | 0.000±0.000 | **0.349±0.063** |
| **PushBlock** | 0.003±0.004 | 0.011±0.003 | 0.025±0.008 | 0.069±0.049 | **0.378±0.022** |
| **DoorOpenInward** | 0.000±0.000 | 0.004±0.004 | 0.007±0.010 | 0.002±0.002 | **0.283±0.386** |
| **ReOrientation** | 0.021±0.004 | 0.028±0.003 | 0.107±0.019 | 0.060±0.027 | **0.149±0.026** |
| **Switch** | 0.000±0.000 | 0.000±0.000 | 0.000±0.000 | 0.000±0.000 | 0.000±0.000 |

### A.2.2 RESULT COMPARISON OF HNS

In our experiments, we introduce the Human Normalized Score (**HNS**) as the second evaluation metrics. HNS assesses a method's performance relative to human-engineered reward functions and is computed as:

$$\text{HNS} = \frac{\text{MTS}_{\text{Method}} - \text{MTS}_{\text{Sparse}}}{\text{MTS}_{\text{Human}} - \text{MTS}_{\text{Sparse}}}. \tag{8}$$

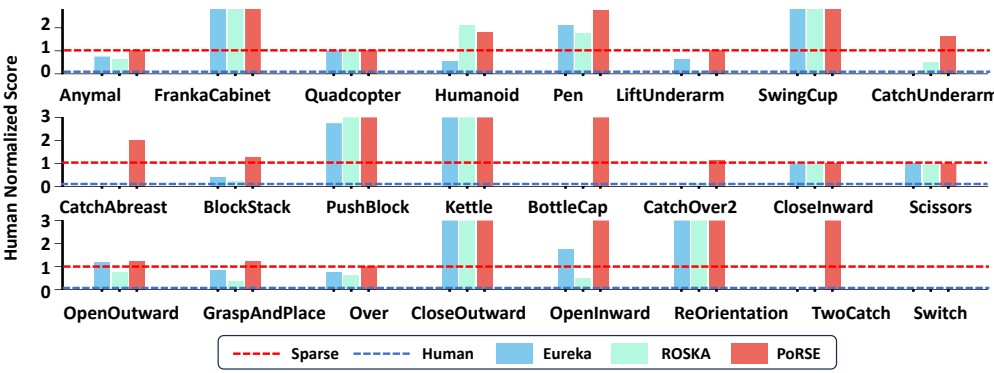

Figure 7: HNS score comparison across 24 robotic tasks. PoRSE significantly outperformed other methods in 23 (all 24) robot skill learning tasks, achieving the best experimental results.

where $MTS_{Method}$ represents the score achieved by a particular method, $MTS_{Sparse}$ denotes the baseline sparse reward, and $MTS_{Human}$ corresponds to expert human performance.

From the experimental results and the corresponding visualization chart, with Human (human-engineered rewards, HNS = 1) and Sparse (sparse reward baseline, HNS = 0) as references, the proposed method (Ours/PoRSE) demonstrates significant advantages in most tasks: On the one hand, in tasks requiring complex operations or fine-grained control (e.g., FrankaCabinet, SwingCup, PushBlock, Kettle), the HNS of Ours far exceeds that of Human (for instance, the HNS of Ours in SwingCup reaches 41.42, which is much higher than Human's 1 and the performance of other methods), demonstrating that it can learn better reward strategies than human-engineered ones and drive policies to achieve performance far beyond expert-level. On the other hand, in relatively simple tasks or those with mature solutions (e.g., Quadcopter, Scissors), the HNS of Ours is comparable to that of Human or other state-of-the-art methods, reflecting its adaptability to different types of tasks. In contrast, comparative methods Eureka and ROSKA even yield negative HNS in tasks like BottleCap, while Ours still achieves positive and excellent performance — this further reflects that Ours is superior in the stability and effectiveness of reward learning, can guide policy learning more efficiently, and possesses stronger capabilities of performance generalization and optimization in various robotic manipulation tasks.

### A.2.3 DETAILED EXPERIMENTAL RESULTS OF LLMCOUNT

We conducted extensive experiments on the LLMCountSarukkai et al. (2024) method on 24 reinforcement learning tasks, with 5 different random seed results collected and averaged for each task. The detailed comparison between LLMCount method and PoRSE method is as follows.

As shown in Tab. 7, the PoRSE method significantly outperforms the LLMCount benchmark method on 23 of the 24 robot skill learning tasks. Specifically, in the Quadruped Robot Motion Control task (Anymal), PoRSE improves the Mean Training Success (MTS) from -1.226 in LLMCount to -0.012, which is close to the expert human-designed level of -0.021. As for the TwoCatch task, which requires precise operation, LLMCount fails completely (MTS=0), while PoRSE achieves a success rate of 0.349 through the dynamic reward weight adjustment mechanism, indicating that it can effectively solve the exploration-exploitation dilemma under sparse rewards. The experimental results comprehensively validate the combined advantages of the PoRSE framework by synergistically optimizing the goal-oriented reward and state-exploration bonus mechanisms.

### A.2.4 MAPPING FUNCTION ABLATION EXPERIMENT AND ANALYSIS

To evaluate the actual contribution of the Mapping Function to the PoRSE framework, we employed an alternative mapping function—simhashTang et al. (2017). This function can directly map similar high-dimensional environmental state information into the same interval, serving as the foundation for count-based rewards. Unlike the Mapping Function in this paper, simhash lacks any semantic

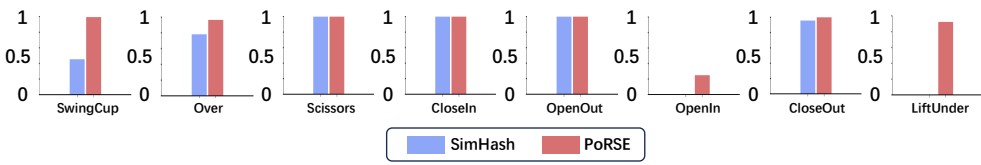

Figure 8: Success rate comparison of SimHash and PoRSE methods

Table 7: MTS Comparison of LLMCount and PoRSE methods, PoRSE method achieved higher MTS on 23 tasks.

| Method | Anymal | Franka | Quadcopter | Humanoid | LiftUnderarm | Pen |
|---|---|---|---|---|---|---|
| LLMCount | -1.226 | 0.706 | -0.092 | 7.737 | 0.649 | 0.412 |
| PoRSE | **-0.012** | **0.957** | **-0.014** | **8.454** | **0.952** | **1.000** |

| Method | SwingCup | CatchUnder | CatchAbreast | BlockStack | PushBlock | Kettle |
|---|---|---|---|---|---|---|
| LLMCount | 0.993 | 0.000 | 0.238 | 0.140 | 0.127 | **1.000** |
| PoRSE | **0.995** | **0.894** | **0.745** | **0.753** | **0.378** | **1.000** |

| Method | BottleCap | CatchOver2 | CloseIn | Scissors | OpenOut | GraspPlace |
|---|---|---|---|---|---|---|
| LLMCount | 0.985 | 0.387 | **1.000** | **1.000** | 0.998 | 0.906 |
| PoRSE | **0.988** | **0.973** | **1.000** | **1.000** | **1.000** | **0.984** |

| Method | Over | CloseOut | OpenIn | ReOrientation | TwoCatch | Switch |
|---|---|---|---|---|---|---|
| LLMCount | 0.594 | 0.905 | 0.025 | 0.098 | 0.000 | 0.000 |
| PoRSE | **0.965** | **1.000** | **0.283** | **0.149** | **0.349** | 0.000 |

Table 8: MTS Comparison of SimHash and PoRSE methods, PoRSE method achieved higher MTS in 8 robot skill learning tasks.

| Method | SwingCup | Over | Scissors | CloseIn | OpenOut | OpenIn | CloseOut | LiftUnder |
|---|---|---|---|---|---|---|---|---|
| SimHash | 0.454 | 0.782 | **1.000** | **1.000** | **1.000** | 0.000 | 0.958 | 0.000 |
| PoRSE | **0.995** | **0.965** | **1.000** | **1.000** | **1.000** | **0.250** | **1.000** | **0.952** |

connection to the task goal and relies solely on the similarity of high-dimensional environmental state data to partition intervals and perform counting. Thus, this comparative experiment is designed to validate the effectiveness of the Mapping Function adopted in this paper.

As shown in Tab. 8, experiments reveal that the SimHash mapping function, devoid of task semantics, underperforms PoRSE across all tasks. Semantic state abstraction via language models markedly enhances exploration efficiency. PoRSE's task-relevant state mapping directs agents to focus on key behavioral features, unlike SimHash's count-based mechanism reliant on high-dimensional state similarity, which struggles to escape local optima. This validates the necessity of task-driven abstract state space design for complex skill learning.

### A.2.5 LLM ABLATION EXPERIMENT AND ANALYSIS

To verify the robustness of our framework to the output quality of large language models (LLMs), we design this ablation study. Specifically, we replace the LLM used in the original method with

a more lightweight GPT-4o-mini model (denoted as PoRSE-GPT-4o-mini) and compare its performance with the original method and other baselines across multiple tasks. The core purpose of this experiment is to investigate whether the proposed co-optimization paradigm can still maintain its effectiveness when the quality of the underlying LLM changes, thereby demonstrating that the advantage of the framework does not rely on a specific high-performance LLM, but rather on its inherent and dynamic optimization mechanism.

Table 9: Performance Comparison of PoRSE-GPT-4o-mini with Baselines on Representative Tasks

| Task | BlockStack | PushBlock | LiftUnderarm | CatchUnderarm |
|---|---|---|---|---|
| Sparse | 0.000 ± 0.001 | 0.003 ± 0.004 | 0.000 ± 0.000 | 0.000 ± 0.000 |
| Human | 0.600 ± 0.229 | 0.011 ± 0.003 | 0.348 ± 0.140 | 0.544 ± 0.250 |
| Eureka | 0.254 ± 0.119 | 0.025 ± 0.008 | 0.739 ± 0.166 | 0.000 ± 0.000 |
| Roska | 0.148 ± 0.154 | 0.069 ± 0.049 | 0.608 ± 0.211 | 0.271 ± 0.370 |
| PoRSE-GPT-4o-min | 0.618 ± 0.165 | 0.360 ± 0.051 | 0.869 ± 0.07 | 0.907 ± 0.015 |
| PoRSE | **0.753 ± 0.210** | **0.378 ± 0.022** | **0.952 ± 0.015** | **0.894 ± 0.038** |

As can be observed from the experimental results (shown in the table below), even when using the less performant GPT-4o-mini model, PoRSE-GPT-4o-mini significantly outperforms all baseline methods except the original PoRSE on the four tasks: BlockStack, PushBlock, LiftUnderarm, and CatchUnderarm. For instance, in the PushBlock task, its performance (0.36±0.051) far exceeds that of Roska (0.069±0.049) and Eureka (0.025±0.008); in the CatchUnderarm task, its success rate even reaches 0.907±0.015. This fully demonstrates that although the initial output quality of the LLM decreases, the strategy can still be guided to converge to a high-performance level through multiple rounds of dynamic optimization within the framework. This result strongly verifies the robustness of the framework to the variability of LLM outputs, and its core value lies in establishing a "goal-exploration-policy" co-optimization paradigm that is not strongly tied to a specific LLM and can continuously self-improve.

A.2.6  AFS ABLATION EXPERIMENT AND ANALYSIS

Table 10: MTS Performance Comparison of PoRSE-AFS-Random and Baseline Methods on Representative Manipulation Tasks (mean ± standard deviation)

| Task | BlockStack | PushBlock | LiftUnderarm |
|---|---|---|---|
| Sparse | 0.000 ± 0.001 | 0.003 ± 0.004 | 0.000 ± 0.000 |
| Human | 0.600 ± 0.229 | 0.011 ± 0.003 | 0.348 ± 0.140 |
| Eureka | 0.254 ± 0.119 | 0.025 ± 0.008 | 0.739 ± 0.166 |
| Roska | 0.148 ± 0.154 | 0.069 ± 0.049 | 0.608 ± 0.211 |
| PoRSE-AFS-Random | 0.680 ± 0.080 | 0.324 ± 0.034 | 0.802 ± 0.029 |
| PoRSE | **0.753 ± 0.210** | **0.378 ± 0.022** | **0.952 ± 0.015** |

To verify the fault tolerance of the PoRSE framework to invalid/inefficient affordance state space (AFS) dimensions generated by large language models (LLMs), as well as the core supporting role of the built-in elimination mechanism in task adaptability, this ablation study designs a control group with "randomly assembled AFS" (denoted as PoRSE-AFS-Random). Although the original PoRSE framework leverages LLMs to generate task-relevant AFS dimensions for guiding efficient exploration, in practical applications, LLMs may generate goal-irrelevant dimensions due to task comprehension biases or lead to degraded AFS quality due to initial design flaws. To simulate this worst-case scenario, the experiment constructs invalid AFS by randomly splicing environmental state variable dimensions. The aim is to explore whether the framework can dynamically screen dimensions valuable for policy optimization through the elimination mechanism, while verifying whether the framework can still maintain performance superior to baselines even when the initial AFS quality is poor—thereby demonstrating its core advantage of strong robustness that does not rely on specific high-quality LLM outputs.

As shown in Table 10, even with the randomly assembled invalid AFS (PoRSE-AFS-Random), its Maximum Training Success (MTS) on three representative tasks (BlockStack, PushBlock, and LiftUnderarm) is still significantly superior to that of baseline methods including Sparse, Human, Eureka, and Roska. Specifically, in the BlockStack task, the MTS of PoRSE-AFS-Random (0.680±0.080) is 2.67 times and 4.59 times that of Eureka (0.254±0.119) and Roska (0.148±0.154), respectively. In the PushBlock task, its performance (0.324±0.034) far exceeds all baselines (the highest-performing baseline, Roska, only achieves 0.069±0.049). In the LiftUnderarm task, its MTS (0.802±0.029) is also better than that of Eureka (0.739±0.166) and Roska (0.608±0.211). Meanwhile, although PoRSE-AFS-Random is slightly inferior to the original PoRSE (which relies on valid AFS generated by LLMs), during the experiment, the framework dynamically eliminates invalid dimensions through the elimination mechanism and adaptively reduces the exploration reward coefficient via the LLM to avoid interfering with policy training. Ultimately, it still maintains significant performance advantages in the scenario of random AFS, fully verifying the framework's fault tolerance to AFS dimension invalidity and the critical role of the elimination mechanism.

### A.2.7 DIVERSITY OF DIMENSIONS IN AFS ABLATION EXPERIMENT AND ANALYSIS

Table 11: Maximum Training Success (MTS) Comparison of PoRSE and PoRSE-Prompt on AFS Dimension Diversity Experiments

| Method | FrankaCabinet | OpenInward | OpenOutward |
|---|---|---|---|
| PoRSE-Prompt | 0.951 ± 0.031 | 0.264 ± 0.044 | 1.000 ± 0.000 |
| PoRSE | **0.957 ± 0.049** | **0.283 ± 0.386** | **1.000 ± 0.000** |

To investigate the regulatory effect of Prompt design on the scope of affordance state space (AFS) dimensions generated by large language models (LLMs), as well as the impact of including dimensions not directly related to the task goal on policy training, this ablation study designs a "Prompt-guided group" (denoted as PoRSE-Prompt). In the original PoRSE framework, the AFS dimensions generated by LLMs are mostly centered on the task goal (e.g., focusing on task-directly relevant dimensions such as "door handle distance" in door control tasks). However, in practical applications, it is necessary to verify two key points: first, whether LLMs can expand the scope of AFS dimensions (to include dimensions not directly related to the task, such as "exploration of the unknown position of the handle") through Prompt guidance; second, whether this dimension diversity will interfere with task completion. The core motivation of the experiment is to clarify the ability of Prompt design to regulate AFS composition, while verifying the framework's compatibility with AFS dimension redundancy—i.e., whether task-irrelevant dimensions will impair task success rate—thereby providing experimental support for the flexibility of AFS design.

As shown in Table 11, there is no significant difference in Maximum Training Success (MTS) between PoRSE-Prompt and the original PoRSE across three representative tasks (FrankaCabinet, OpenInward, and OpenOutward). Specifically, in the FrankaCabinet task, the MTS of PoRSE-Prompt (0.951±0.031) only shows a slight decrease of 0.6 percentage points compared to the original PoRSE (0.957±0.049). In the OpenInward task, its MTS (0.264±0.044) is close to that of the original PoRSE (0.283±0.386), with weaker data volatility. Notably, in the OpenOutward task, both methods achieve a 100% success rate (1.0±0.0). These results indicate that after adding guiding statements (e.g., "explore the unknown position of the handle") to the Prompt, the AFS generated by LLMs includes dimensions not directly related to the task; however, these redundant dimensions do not interfere with policy training, as reflected by the insignificant change in task success rate. This finding not only verifies the regulatory role of Prompt design in the scope of AFS dimensions but also proves that the PoRSE framework has good compatibility with AFS dimension diversity, providing an experimental basis for the flexible design of AFS in subsequent studies.

### A.2.8 BIN COUNTS ABLATION EXPERIMENT AND ANALYSIS

To clarify the rationality of the "bin count" parameter in the exploration bonus generation process and address potential ambiguity regarding this parameter, this ablation study conducts verification on the bin count during Affordance State Space (AFS) discretization. As shown in the main text,

Table 12: Maximum Training Success (MTS) Comparison of PoRSE with Different Bin Counts for Exploration Bonuses

| Method | BlockStack | PushBlock | LiftUnderarm |
|---|---|---|---|
| PoRSE (1000 bins) | 0.753 ± 0.210 | **0.378 ± 0.022** | **0.952 ± 0.015** |
| PoRSE (2000 bins) | 0.708 ± 0.056 | 0.335 ± 0.103 | 0.950 ± 0.023 |
| PoRSE (3000 bins) | **0.763 ± 0.266** | 0.321 ± 0.027 | 0.922 ± 0.079 |

the calculation of the exploration bonus ($R_e$) relies on converting the continuous AFS into discrete states via the discretization function $D(S_o)$, and the bin count directly determines the discretization granularity. Although the default bin count was explicitly stated as 1000 in the previous supplementary materials, the impact of this parameter on performance had not been verified. Three bin counts (1000, 2000, and 3000) were selected as variables in the experiment, with core motivations as follows: on the one hand, to confirm whether the bin count significantly affects the task success rate (Maximum Training Success, MTS) and avoid result biases caused by subjective parameter selection; on the other hand, to provide experimental evidence for adopting 1000 bins as the default setting, ensuring the scientific validity and reproducibility of the framework parameters.

As shown in Table 12, different bin counts have no significant impact on the MTS of PoRSE across the three tasks (BlockStack, PushBlock, and LiftUnderarm). Specifically, in the BlockStack task, the MTS values of PoRSE with 1000 bins (0.753±0.210), 2000 bins (0.708±0.056), and 3000 bins (0.763±0.266) fluctuate within the range of 0.708 to 0.763, with no obvious performance superiority or inferiority. In the PushBlock task, although PoRSE with 1000 bins achieves the optimal performance (0.378±0.022), the performance gaps between 2000 bins (0.335±0.103) and 3000 bins (0.321±0.027) are all within 0.05, and there is no statistical significance in these differences. In the LiftUnderarm task, the MTS values of all three settings maintain a high level of 0.922 to 0.952, with only PoRSE with 3000 bins (0.922±0.079) showing slight fluctuations due to excessively fine binning granularity. This result confirms that the bin count within the range of 1000 to 3000 does not exert a critical impact on policy training effectiveness. Therefore, selecting 1000 bins as the default setting not only meets the requirement of AFS discretization but also avoids computational resource waste caused by excessive bins, achieving a balance between performance and efficiency.

A.2.9 NORMILIZATION ABLATION EXPERIMENT AND ANALYSIS

Table 13: Maximum Training Success (MTS) Comparison of PoRSE with Different Reward Coefficient Sum Scales

| Method | BlockStack | PushBlock | LiftUnderarm |
|---|---|---|---|
| PoRSE (Normalized-to-1) | 0.744 ± 0.201 | 0.353 ± 0.077 | **0.957 ± 0.020** |
| PoRSE | **0.753 ± 0.210** | **0.378 ± 0.022** | 0.952 ± 0.015 |

To explain the rationality of the design where "the sum of coefficients is fixed to 2" in the total reward formula and verify the impact of reward coefficient scale selection on performance, this ablation study designs a control group with "coefficient sum normalized to 1" (denoted as PoRSE-Normalized-to-1). As shown in the main text, the total reward formula of the PoRSE framework is $R_{total}^k = \beta \cdot R_e^k + (2 - \beta) \cdot R_g^k$ (where $\beta$ is the weight of the exploration bonus $R_e$, $2 - \beta$ is the weight of the goal-oriented reward $R_g$, and the sum of the coefficients is fixed to 2). This design is essentially a matter of scale selection. The experiment has two core motivations: first, to verify whether there is a significant difference in Maximum Training Success (MTS) when the sum of coefficients is "1" versus "2", eliminating the risk of scale dependence; second, from the perspective of optimization efficiency, to prove that "independent optimization of two coefficients (including a scale variable)" is mathematically equivalent to "single-coefficient optimization (fixed scale, only optimizing $\beta$)". If the scale has no significant impact, the scale variable can be omitted and only a single coefficient optimized, greatly simplifying the framework's optimization process.

As shown in Table 13, the impact of the coefficient sum being 2 (original PoRSE) versus 1 (PoRSE-Normalized-to-1) on MTS is minimal. Specifically, in the BlockStack task, their MTS values are 0.753±0.210 and 0.744±0.201 respectively, with a difference of only 0.009 and no substantial performance gap. In the PushBlock task, although the original PoRSE (0.378±0.022) is slightly superior to the normalized group (0.353±0.077), the gap is within 0.03 and there is no statistical significance. In the LiftUnderarm task, the normalized group (0.957±0.020) even slightly outperforms the original PoRSE (0.952±0.015). This result fully confirms that the sum of reward coefficients, within the scale range of "1" or "2", does not exert a critical impact on policy training effectiveness. Therefore, the design of the original PoRSE—"fixing the sum to 2 and only optimizing the single coefficient $\beta$"—not only ensures performance stability but also avoids computational redundancy caused by multi-variable optimization, achieving a balance between performance and optimization efficiency.

### A.2.10 COMPONENTS OF PORSE ABLATION EXPERIMENT AND ANALYSIS

To verify the necessity of the three core components of the PoRSE framework—goal-oriented reward, exploration bonus, and reward-policy co-optimization—and to demonstrate that the framework is not a simple superposition of existing modules, this ablation study systematically compares PoRSE with representative methods in the field (Eureka, ROSKA, ROSKA+CE) in terms of component composition and performance. As stated in the main text, the core innovation of PoRSE lies in using "dynamic policy feedback" as a link to integrate the three components into an organic whole, rather than treating them as isolated parts. In contrast, existing methods either lack exploration bonuses (Eureka, ROSKA) or only achieve simple component aggregation through "traditional count-based exploration (CE) + ROSKA" (ROSKA+CE), with neither realizing inter-component synergy. Three tasks—BlockStack, PushBlock, and LiftUnderarm—covering both "complex manipulation" and "high-efficiency exploration requirements" were selected for the experiment. The core objectives are: to prove the simplicity (only three core components) and necessity of PoRSE's component composition through performance differences caused by the presence or absence of components; and to highlight the critical role of "component synergy" (rather than "simple superposition") in achieving performance breakthroughs by comparing with ROSKA+CE.

Table 14: Component Composition Comparison of PoRSE with State-of-the-Art Methods

| Methods | Goal Reward | Exploration Bonus | Reward-Policy Co-optimization |
|---|---|---|---|
| Eureka | $\checkmark$ | $\times$ | $\times$ |
| ROSKA | $\checkmark$ | $\times$ | $\checkmark$ |
| ROSKA+CE | $\checkmark$ | $\checkmark$ | $\checkmark$ |
| PoRSE | $\checkmark$ | $\checkmark$ | $\checkmark$ |

As seen from the component composition (Table 14), PoRSE adopts a minimalist "three-component architecture," with each component being indispensable for efficient skill learning. Firstly, the goal-oriented reward serves as the foundation for task guidance—all compared methods include this component (marked with $\checkmark$), and its absence would result in no clear optimization direction, a consensus in robotic RL tasks. However, Eureka, which relies solely on goal-oriented rewards (lacking exploration and co-optimization), only achieves an MTS of 0.025±0.008 in the PushBlock task, struggling to escape local optima. Secondly, the exploration bonus is key to efficient exploration—ROSKA, which lacks this component, achieves an MTS of 0.608±0.211 in the LiftUnderarm task, significantly lower than PoRSE (0.952±0.015) which includes an exploration bonus. Although ROSKA+CE adds an exploration bonus, it adopts traditional count-based exploration irrelevant to the task, leading to performance far inferior to PoRSE. Thirdly, reward-policy co-optimization is central to performance enhancement—Eureka, without co-optimization, cannot dynamically adjust the adaptability between rewards and policies, resulting in the worst performance in complex tasks (e.g., BlockStack). These three components together form a minimalist yet necessary component system for PoRSE; the absence of any one component makes efficient skill learning unattainable.

The core advantage of PoRSE lies not in the "presence or absence" of components, but in the mutually reinforcing effect formed between components through "dynamic policy feedback"—an effect

Table 15: MTS Performance Comparison of PoRSE with State-of-the-Art Methods (mean ± standard deviation)

| Methods | BlockStack | PushBlock | LiftUnderarm |
|---|---|---|---|
| Eureka | 0.254 ± 0.119 | 0.025 ± 0.008 | 0.739 ± 0.166 |
| ROSKA | 0.148 ± 0.154 | 0.069 ± 0.049 | 0.608 ± 0.211 |
| ROSKA+CE | 0.217 ± 0.056 | 0.129 ± 0.040 | 0.664 ± 0.072 |
| PoRSE | **0.753 ± 0.210** | **0.378 ± 0.022** | **0.952 ± 0.015** |

clearly reflected in the experimental results. On the one hand, goal-oriented rewards anchor the direction for exploration bonuses: PoRSE's exploration bonus is based on task-relevant Affordance State Space (AFS) generated by LLMs, rather than the "goal-agnostic count-based exploration" of ROSKA+CE. This focuses exploration resources on task-critical dimensions (e.g., "block stacking height" in BlockStack), avoiding ineffective exploration. On the other hand, exploration bonuses expand the optimization space for goal-oriented rewards: in the PushBlock task, PoRSE discovers the "optimal state for bimanual collaborative pushing" through exploration bonuses, improving the optimization efficiency of goal-oriented rewards. Its MTS (0.378±0.022) is 2.93 times that of ROSKA+CE (0.129±0.040). Meanwhile, reward-policy co-optimization (via the IPG process) dynamically balances the two components: it adjusts the $\beta$ coefficient based on real-time policy performance, enabling adaptive matching between goal-oriented behavior and exploration as the policy evolves. This positive cycle of "goal-anchored exploration, exploration feeding back to goal-oriented rewards, and co-optimization maintaining balance" allows PoRSE to significantly outperform existing methods relying on simple component superposition across all tasks, fully verifying the irreplaceability of component synergy.

### A.2.11 ABLATION EXPERIMENTAL DROP RESULTS

Table 14 16 summarizes the percentage performance drop of each ablation relative to the full PoRSE configuration across six representative tasks. The largest degradations arise when the fusion mechanism ($\theta_{\text{fusion}}$) is removed, with particularly severe declines on locomotion tasks such as Anmal ($\downarrow$ 2783.33%) and Quadcopter ($\downarrow$ 142.86%), and a marked decrease on the long horizon sparse manipulation task TwoCatch ($\downarrow$ 95.13%). This pattern indicates that policy grounded fusion is central to PoRSE, since the dynamic blending of goal oriented rewards ($R_g$) and exploration bonuses ($R_e$) stabilizes learning and sustains final performance. Eliminating either ($R_g$) or ($R_e$) also harms outcomes across domains. On Anmal the absence of ($R_g$) leads to ($\downarrow$ 966.67%) and the absence of ($R_e$) to ($\downarrow$ 708.33%), which highlights the complementary roles of goal shaping and count based exploration provided by the AFS space and the IPG loop.

The ratio related terms ($R_{\text{ratio}}$) and ($\theta_{\text{ratio}}$) have a milder yet meaningful effect. Their removal produces moderate declines on manipulation tasks, for example BlockStack ($\downarrow$ 60.70%) without ($R_{\text{ratio}}$) and PushBlock ($\downarrow$ 23.58%), suggesting that ratio based normalization helps calibrate reward magnitudes and maintain comparability of AFS counts when scenes involve multiple objects or changing contact phases. Overall, the ablations support three conclusions that are consistent with the design goals of PoRSE. The policy grounded fusion is the dominant contributor to robustness and efficiency, both ($R_g$) and ($R_e$) are necessary to realize the intended synergy, and the ratio based components further stabilize learning in compositional settings.

### A.2.12 REVOLVE-AUTO EXPERIMENT CONFIGURATION AND ANALYSIS

To ensure fair comparison with PoRSE, the Revolve-AutoHazra et al. (2024) variant was aligned with PoRSE's core experimental settings: it generates 6 reward functions per iterative round and runs for 5 total iterative rounds. Table. 17 (Experimental comparison of PoRSE with SOTA frameworks including REvolve) supplements the experimental results of the REvolve framework on four representative tasks (Humanoid, Liftunderarm, BlockStack, TwoCatch) to further verify PoRSE's superiority over state-of-the-art (SOTA) reward-design frameworks. For fair comparison, we made

Table 16: Performance drop (%) of each ablation compared with the full PoRSE framework.

| Method | Anymal | Quadcopter | Franka | BlockStack | PushBlock | TwoCatch |
|---|---|---|---|---|---|---|
| PoRSE w/o $R_g$ | 966.7% | 185.7% | 7.740% | 56.44% | 7.230% | 45.56% |
| PoRSE w/o $R_e$ | 708.3% | 171.4% | 4.700% | 47.80% | 3.770% | 44.70% |
| PoRSE w/o $\theta_{\text{fusion}}$ | 2783% | 142.8% | 29.89% | 19.92% | 25.79% | 95.13% |
| PoRSE w/o $R_{\text{ratio}}$ | 450.0% | 35.71% | 1.150% | 60.70% | 23.58% | 14.90% |
| PoRSE w/o $\theta_{\text{ratio}}$ | 66.67% | 7.140% | 1.780% | 21.64% | 2.830% | 20.92% |

Table 17: MTS (mean $\pm$ std) across four tasks.

| Method | Humanoid | Liftunderarm | BlockStack | TwoCatch |
|---|---|---|---|---|
| Eureka | $6.534 \pm 1.933$ | $0.739 \pm 0.166$ | $0.254 \pm 0.119$ | $0.001 \pm 0.001$ |
| ReVolve | $7.894 \pm 0.635$ | $0.814 \pm 0.079$ | $0.342 \pm 0.054$ | $0.000 \pm 0.000$ |
| PoRSE | $8.454 \pm 0.343$ | $0.952 \pm 0.015$ | $0.753 \pm 0.210$ | $0.349 \pm 0.063$ |

key adjustments to the original REvolve setup: since the original REvolve relies on human participation in its reward optimization loop, we removed human intervention and migrated the complete Auto-REvolve framework into the Eureka pipeline (denoted as Revolve-Auto). This adaptation retains the evolutionary processes and prompt information exactly as described in the original REvolve paper, while aligning evaluation metrics (Maximum Training Success, MTS) with those used for PoRSE.

The table includes three methods for comparison: Eureka (LLM-driven reward generation), Revolve-Auto (human-free REvolve adaptation), and PoRSE (our proposed framework). The results show that PoRSE comprehensively outperforms the other two methods across all tasks: In the Humanoid task, PoRSE achieves an MTS of 8.454±0.343, surpassing Revolve-Auto (7.894±0.635) and Eureka (6.534±1.933); in Liftunderarm, PoRSE's MTS (0.952±0.015) outperforms Revolve-Auto (0.814±0.079) and Eureka (0.739±0.166) by a notable margin; in BlockStack, PoRSE's performance (0.753±0.210) is more than twice that of Eureka (0.254±0.119) and nearly double that of Revolve-Auto (0.342±0.054); most notably, in the highly challenging TwoCatch task (requiring synchronized bimanual coordination), PoRSE achieves a breakthrough MTS of 0.349±0.063, while both Revolve-Auto and Eureka fail to make meaningful progress (MTS = 0 and 0.001±0.001, respectively).

This performance gap stems from core design differences: Revolve-Auto, like the original REvolve, focuses primarily on evolutionary goal-oriented reward design but lacks task-aware exploration mechanisms, limiting its effectiveness in high-dimensional or sparse-reward tasks. In contrast, PoRSE's In-Policy-Improvement Grounding (IPG) mechanism dynamically adjusts rewards and exploration bonuses based on real-time policy performance, while its Affordance State Space (AFS) enables task-relevant exploration—these synergistic designs drive sustained policy optimization.

### A.2.13 INTUITION FOR LLM-CHOSEN

This section elaborates on the reasoning underlying the large language model (LLM)-predicted policy fusion coefficient $\alpha$ and reward weight coefficient $\beta$ in the PoRSE framework, as well as the numerical evolutionary trends of these coefficients across training iterations.

The intuition for leveraging LLMs to predict $\alpha$ and $\beta$ originates from the model's inherent reasoning and data analysis capabilities—strengths validated in prior work focused on LLM-driven decision-making [1], self-composed reasoning structures [2], and enhanced mathematical reasoning via code integration [3]. In the PoRSE framework, we input time-series training data (e.g., Maximum Training Success (MTS) over epochs) of the current policy to the LLM, allowing the model to assess the policy's training state (e.g., whether it is trapped in a local optimum or nearing convergence). Rather than generating a single numerical value for $\alpha$ and $\beta$, the LLM produces a set of candidate

outputs; this multi-candidate design mitigates biases from isolated predictions, thereby improving the stability of coefficient selection. The full prompt template guiding LLM predictions of $\alpha$ and $\beta$ is documented later in this section.

To characterize the evolutionary patterns of $\alpha$ and $\beta$ (as noted in the main text L302–304), we collected their values across training iterations for representative robotic tasks. Since PoRSE alternates between optimizing $\alpha$ (policy fusion coefficient) and $\beta$ (reward weight coefficient) across iterations, we tracked 6 total iterations: $\alpha$ data were gathered at Iterations 1, 3, 5, while $\beta$ data were collected at Iterations 0, 2, 4. The collected values for each task are presented in Table 18:

| Tasks | Iter1 $\alpha$ | Iter3 $\alpha$ | Iter5 $\alpha$ | Iter0 $\beta$ | Iter2 $\beta$ | Iter4 $\beta$ |
|---|---|---|---|---|---|---|
| Humanoid | 0.142 | 0.473 | 0.815 | 1.725 | 1.363 | 0.784 |
| Anymal | 0.137 | 0.468 | 0.809 | 1.572 | 1.191 | 0.793 |
| Quadcopter | 0.214 | 0.562 | 0.863 | 1.734 | 1.462 | 0.831 |
| BlockStack | 0.218 | 0.571 | 0.872 | 1.673 | 1.184 | 0.842 |
| Liftunderarm | 0.275 | 0.644 | 0.893 | 1.845 | 1.763 | 0.872 |
| CatchAbreast | 0.281 | 0.652 | 0.901 | 1.881 | 1.282 | 1.064 |
| PushBlock | 0.332 | 0.715 | 0.934 | 1.823 | 1.371 | 0.893 |
| TwoCatchUnderarm | 0.341 | 0.723 | 0.942 | 1.942 | 1.684 | 1.085 |

Table 18: Numerical values of $\alpha$ and $\beta$ across training iterations for representative tasks.

Analysis of Table 18 reveals consistent evolutionary trends across all tasks: $\alpha$ values increase monotonically across iterations, indicating that PoRSE increasingly prioritizes inheriting parameters from the best-performing historical policy as training progresses—strengthening policy consistency and leveraging prior effective learning progress. In contrast, $\beta$ values decrease across iterations, reflecting a strategic adjustment: early stages use higher $\beta$ to emphasize exploration bonuses (aiding escape from local optima), while later stages reduce $\beta$ to prioritize goal-oriented rewards (refining task completion performance).

## A.3 EXPERIMENTAL SETUP

In this section, we present the detailed experimental parameter setup for all methods in this paper, provide an introduction to the IsaacGym tasks, outline the computing infrastructure, and detail the calculation of total training epochs (TTE). The PoRSE code will be released at the following URL: `https://anonymous.4open.science/r/PoRSE-1C5E`.

### A.3.1 DETAILED EXPERIMENT PARAMETER SETTING

Table 19: Comparison of key parameters among different methods in the paper.

| Method | Iterations | Reward Functions per Iteration | Total Training Epochs (TTE) | Max Policy Training Epochs per Iteration |
|---|---|---|---|---|
| PoRSE | 5 | 6 | 90,000 | 3,000 |
| ROSKA | 5 | 6 | 90,000 | 3,000 |
| Eureka | 5 | 6 | 90,000 | 3,000 |
| LLMCount | 1 | 6 | 90,000 | 15000 |

The specific configurations for all methods in this paper are as follows:

**ROSKA:** The ROSKAHuang et al. (2025) method also conducts 5-iteration optimization, generating 6 reward functions in each iteration and training the corresponding policies in parallel. To ensure fairness in computational resources, we increased the number of Bayesian optimization samples from the default 12 to 14, while other parameters remain as per the original paper.

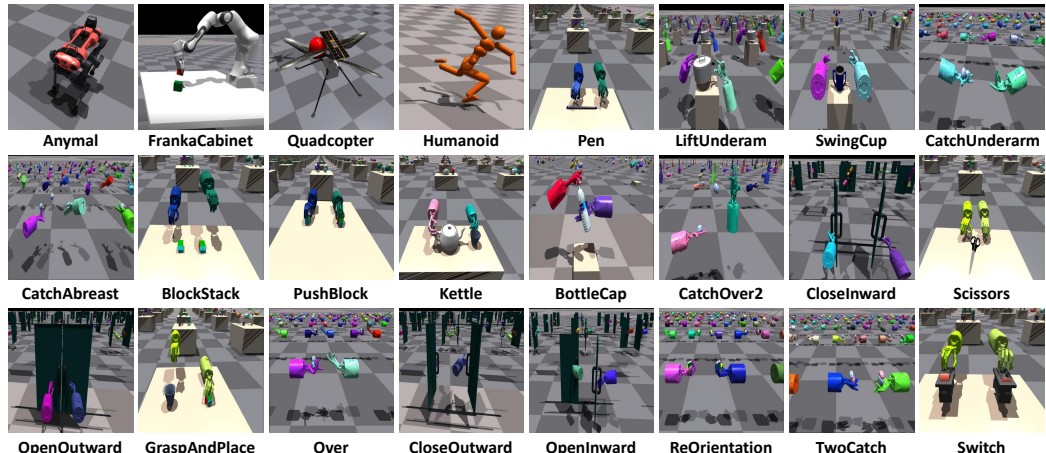

Figure 9: The 24 reinforcement learning task example images selected in the experimental section of this paper.

**Euereka:** The EurekaMa et al. (2023) method employs the same 5-iteration structure, generating 6 reward functions per iteration and allocating 3000 training epochs to each policy for thorough optimization.

**LLMCount:** The LLMCountSarukkai et al. (2024) method used in the ablation study generates 6 progress functions zero-shot through the large language model, with each function allocated 15,000 training epochs to ensure balanced computational resources.

**Human and Sparse:** For the **Human**Makoviychuk et al. (2021) and **Sparse**Ma et al. (2023) methods, we used their default experimental configurations, conducting policy training for 3000 epochs.

**PoRSE:** The detailed parameter configuration of our method has been explained in section 1.

### A.3.2 CALCULATION OF TTE

We have detailed statistics on the TTE for the different methods discussed in this paper. For each task, we use epoch as the statistical unit to measure each method. For the same reinforcement learning task, the policy will interact with the environment in the same epoch for the same number of environment steps. We count the total number of epochs spent on each method during the iteration process to ensure fairness in computing resources.

All compared methods—Eureka, ROSKA, LLMCount, and PoRSE—are designed to use a similar computational budget, each totaling 90,000 training epochs. While their approaches differ in iteration counts, reward function generation, and progressive training schemes, they maintain computational parity to ensure a fair comparison in evaluation.

### A.3.3 COMPUTING PLATFORM

All experiments in this study were conducted on a computer system running Ubuntu 22.04.4 LTS, equipped with eight RTX 4090 GPUs. Since the training time required for training on different tasks is different, for intuitive comparison, we use the total training epochs in each task for comparison. The detailed comparison of Total Training Epochs(TTE) calculations between Eureka, ROSKA, and PoRSE methods is as follows.

### A.3.4 BENCHMARK DESCRIPTION

In this paper, our experiments involve 24 robot skill learning tasks, including 4 tasks from IsaacGym and 20 tasks from Bi-DexHands. The environment task names, task description, and task fitness function F of the 24 tasks are shown in the table 20.

Table 20: The introduction to 24 tasks in the experiments of this paper.

| **IsaacGym Environments** |
| --- |
| **Environment Name(obs dim, action dim)** |
| **Task description** |
| **Task fitness function** $F$ |
| Quadcopter (21, 12) |
| To make the quadcopter reach and hover near a fixed position |
| `-cur_dist` |
| FrankaCabinet (23, 9) |
| To open the cabinet door |
| `1[cabinet_pos > 0.39]` |
| Anymal (48, 12) |
| To make the quadruped follow randomly chosen $x, y$, and yaw target velocities |
| `-(linvel_error + angvel_error)` |
| Humanoid (108, 21) |
| To make the humanoid run as fast as possible |
| `cur_dist - prev_dist` |
| Over (398, 40) |
| This class corresponds to the HandOver task. This environment consists of two shadow hands with palms facing up, opposite each other, and an object that needs to be passed. In the beginning, the object will fall randomly in the area of the shadow hand on the right side. Then the hand holds the object and passes the object to the other hand. Note that the base of the hand is fixed. More importantly, the hand which holds the object initially can not directly touch the target, nor can it directly roll the object to the other hand, so the object must be thrown up and stays in the air in the process |
| `1[dist < 0.03]` |
| DoorCloseInward (417, 52) |
| This class corresponds to the DoorCloseInward task. This environment requires a closed door to be opened and the door can only be pushed outward or initially open inward. Both these two environments only need to do the push behavior, so it is relatively simple. |
| `1[door_handle_dist < 0.5]` |
| DoorCloseOutward (417, 52) |
| This class corresponds to the DoorCloseOutward task. This environment also requires a closed door to be opened and the door can only be pushed inward or initially open outward, but because they can't complete the task by simply pushing, they need to catch the handle by hand and then open or close it, so it is relatively difficult. |
| `1[door_handle_dist < 0.5]` |
| DoorOpenInward (417, 52) |
| This class corresponds to the DoorOpenInward task. This environment also requires an opened door to be closed and the door can only be pushed inward or initially open outward, but because they can't complete the task by simply pushing, they need to catch the handle by hand and then open or close it, so it is relatively difficult. |
| `1[door_handle_dist > 0.5]` |

DoorOpenOutward (417, 52)

This class corresponds to the DoorOpenOutward task. This environment requires an opened door to be closed and the door can only be pushed outward or initially open inward. Both these two environments only need to do the push behavior, so it is relatively simple.

```
1[door_handle_dist < 0.5]
```

Scissors (417, 52)

This class corresponds to the Scissors task. This environment involves two hands and scissors; we need to use two hands to open the scissors.

```
1[dof_pos > -0.3]
```

SwingCup (417, 52)

This class corresponds to the SwingCup task. This environment involves two hands and a dual-handled cup; we need to use two hands to hold and swing the cup together.

```
1[rot_dist < 0.785]
```

Switch (417, 52)

This class corresponds to the Switch task. This environment involves dual hands and a bottle; we need to use dual hand fingers to press the desired button.

```
1[1.4 - (left_switch_z + right_switch_z) > 0.05]
```

Kettle (417, 52)

This class corresponds to the PourWater task. This environment involves two hands, a kettle, and a bucket; we need to hold the kettle with one hand and the bucket with the other hand, and pour the water from the kettle into the bucket. In the practice task in Isaac Gym, we use many small balls to simulate the water.

```
1[|bucket - kettle_spout| < 0.05]
```

LiftUnderarm (417, 52)

This class corresponds to the LiftUnderarm task. This environment requires grasping the pot handle with two hands and lifting the pot to the designated position. This environment is designed to simulate the scene of lift in daily life and is a practical skill.

```
1[dist < 0.05]
```

Pen (417, 52)

This class corresponds to the Open Pen Cap task. This environment involves two hands and a pen; we need to use two hands to open the pen cap.

```
1[5 * |pen_cap - pen_body| > 1.5]
```

Bottle Cap (422, 52)

This class corresponds to the Bottle Cap task. This environment involves two hands and a bottle; we need to hold the bottle with one hand and open the bottle cap with the other hand. This skill requires the cooperation of two hands to ensure that the cap does not fall.

```
1[dist > 0.03]
```

CatchAbreast (422, 52)

This class corresponds to the Catch Abreast task. This environment consists of two shadow hands placed side by side in the same direction and an object that needs to be passed. Compared with the previous environment which is more like passing objects between the hands of two people, this environment is designed to simulate the two hands of the same person passing objects, requiring more hand translation and rotation techniques.

```
1[dist < 0.03]
```

CatchOver2Underarm (422, 52)

This class corresponds to the Over2Underarm task. This environment is similar to Catch Underarm, but with an object in each hand and the corresponding goal on the other hand. Therefore, the environment requires two objects to be thrown into the other hand simultaneously, which demands higher manipulation techniques than single-object environments.

```
1[dist < 0.03]
```

CatchUnderarm (422, 52)

This class corresponds to the Catch Underarm task. In this task, two shadow hands with palms facing upwards are controlled to pass an object from one palm to the other. The difficulty is increased by unfreezing the hands' translation and rotation degrees of freedom in the action space.

```
1[dist < 0.03]
```

ReOrientation (422, 40)

This class corresponds to the ReOrientation task. This environment involves two hands and two objects. Each hand holds an object and we need to reorient the object to the target orientation.

```
1[rot_dist < 0.1]
```

GraspAndPlace (425, 52)

This class corresponds to the GraspAndPlace task. This environment consists of dual hands, an object, and a bucket. The task requires picking up the object and placing it into the bucket.

```
1[|block - bucket| < 0.2]
```

BlockStack (428, 52)

This class corresponds to the Block Stack task. This environment involves dual hands and two blocks; we need to stack the blocks into a stable tower structure.

```
1[goal_dist_1 < 0.07 and goal_dist_2 < 0.07 and z_dist_1 < 0.05]
```

PushBlock (428, 52)

This class corresponds to the PushBlock task. This environment involves two hands and two blocks; we need to use both hands to reach and push each block to its designated goal position simultaneously.

```
1[left_dist <= 0.1 and right_dist <= 0.1]
```

TwoCatchUnderarm (446, 52)

This class corresponds to the TwoCatchUnderarm task. This environment extends the Catch Underarm task with dual-object manipulation: each hand holds an object and must throw it to the opposite hand's target position, requiring synchronized bimanual coordination.

```
1[goal_dist_1 + goal_dist_2 < 0.06]
```

A.3.5 ENVIRONMENT STEP AND PPO UPATE COUNTS

Table 21 (Training budget and compute parity across tasks) is designed to verify the consistency of training resources and computational costs across PoRSE and baseline methods (Eureka, ROSKA), ensuring the fairness of training efficiency comparisons. The table includes 5 representative robotic tasks (Humanoid, FrankaCabinet, Anymal, Quadcopter, Liftunderarm) and the average metrics across all tasks, with columns covering key training budget indicators and GPU time costs.

In terms of training budget uniformity: The "Total epochs" column shows that all tasks maintain a fixed total of 90,000 training epochs for all methods, which aligns with the earlier statement that "PoRSE incorporates mutation and elimination mechanisms but still maintains the same total training epochs as other baselines." For "Total env steps" and "Total PPO updates" (core indicators of training resource input), 4 tasks (Humanoid, FrankaCabinet, Anymal, Quadcopter) share identical values (1,440,000 env steps and 7,200,000 PPO updates), while Liftunderarm uses half the steps

Table 21: Training budget and compute parity across tasks.

| Tasks | Total env steps | Total PPO updates | Total epochs | GPU time (Eureka) | GPU time (ROSKA) | GPU time (PoRSE) |
|---|---|---|---|---|---|---|
| Humanoid | 1,440,000 | 7,200,000 | 90,000 | 7.0 | 7.1 | 7.0 |
| FrankaCabinet | 1,440,000 | 7,200,000 | 90,000 | 4.5 | 4.4 | 4.6 |
| Anymal | 1,440,000 | 7,200,000 | 90,000 | 3.5 | 3.5 | 3.6 |
| Quadcopter | 1,440,000 | 7,200,000 | 90,000 | 3.5 | 3.4 | 3.4 |
| Liftunderarm | 720,000 | 3,600,000 | 90,000 | 5.6 | 5.6 | 5.6 |
| All tasks | – | – | – | 5.5 | 5.7 | 5.8 |

(720,000) and updates (3,600,000) — this difference is task-specific (due to Liftunderarm's lower state dimensionality) and applied uniformly to all methods, ensuring no bias in resource allocation.

In terms of computational cost parity: The "GPU time [h]" columns for Eureka, ROSKA, and PoRSE demonstrate minimal differences across tasks. For example, in the Humanoid task, GPU times are 7.0h (Eureka), 7.1h (ROSKA), and 7.0h (PoRSE); in FrankaCabinet, they are 4.5h, 4.4h, and 4.6h respectively. Even at the average across all tasks, the gap remains negligible (5.5h for Eureka, 5.7h for ROSKA, 5.8h for PoRSE). This confirms that PoRSE does not incur additional computational overhead despite its mutation and elimination mechanisms. Combined with the earlier note that "PoRSE's algorithmic complexity scales linearly with the number of iterations $(O(n))$", the table further validates that PoRSE achieves superior performance without sacrificing training efficiency.

## A.4 DISCUSSION

### A.4.1 DISCUSSION ON DEPLOYMENT

The deployment of the PoRSE framework on physical robots is centered on the core design principle of "decoupling of state representation in the training phase and decision-making in the deployment phase," ensuring the framework can adapt to physical systems without relying on structured state inputs from simulation environments. The Affordance State Space — a core component of the framework — functions exclusively in the training phase, where it is used to construct task-relevant exploration bonus models and low-dimensional state representations to facilitate efficient policy learning. When the policy is deployed on physical robots, however, it can directly take raw sensor data from the robot as inputs for decision-making; typical examples of such raw sensor data include RGB-D images captured by vision cameras and torque and angle information from joint proprioceptive sensors. Discrepancies in dynamics and sensor noise between simulation and real-world environments can be addressed through transfer learning: a specific approach for this adaptation is to fine-tune the trained policy with a small amount of physically collected data. For core AFS features potentially required in real-world scenarios, examples of these features include relative distances between objects and grasping posture angles, and mature existing perception technologies offer efficient solutions to obtain them. For instance, intel RealSense series depth cameras can be used to accurately estimate the spatial positions of objects, while tactile sensors at the robot end or joint torque feedback can infer grasping stability — this means no additional complex feature extraction modules need to be designed. If real-time reward adjustment is required during physical deployment, a lightweight AFS can also be dynamically constructed based on the aforementioned perception data. Furthermore, to enhance the framework's adaptability to unstructured real-world scenarios, common cases of such scenarios include multi-object manipulation in cluttered environments and object occlusion scenarios. Lightweight vision-language models can be integrated into the existing framework pipeline in future work. Representative models of these lightweight vision-language models are CLIP and Grounding DINO, and their integration enables the system to parse task semantics from real-time visual observations of physical robots. This parsing process helps generate reward rules and AFS dimensions that better align with real-world needs, thereby laying a technical foundation for the application of the PoRSE framework on various physical robots. Examples of these physical robots include robotic arms and dexterous hands.

A.4.2 DISCUSSION ON REPRODUCIBILITY

The reproducibility of the PoRSE framework is ensured through multi-dimensional design and information transparency, providing a clear pathway for the verification and extension of subsequent research. At the experimental level, the framework achieves stable performance improvements across 24 tasks covering dexterous manipulation and legged locomotion. The effectiveness of core mechanisms, such as AFS construction and IPG dynamic optimization, has been validated through consistent experimental results, avoiding reproducibility biases caused by task specificity. The settings of key parameters are clearly justified: for example, AFS's default number of bins is set to 1000, this number is used for exploration bonus calculation, and ablation experiments have confirmed that 2000 or 3000 bins have no significant impact on performance; configurations like the number of IPG iterations and the number of reward functions per iteration are detailed in the supplementary materials, and parameter sensitivity analysis results are also provided alongside to guide adaptive adjustments under different scenarios. At the implementation level, the framework will open-source complete code and experimental configuration files, including standardized prompt templates for interactions between LLMs and the framework—such templates require no task-specific tuning, and only task goal descriptions and environmental state variable codes need to be modified—setup scripts for simulation environments like Isaac Gym and Bi-DexHands, and complete process code for policy training and evaluation. This ensures researchers can quickly reproduce the experimental environment. Meanwhile, to address LLM dependency, the framework has verified that lightweight models like GPT-4o-mini can replace the original LLM while maintaining performance advantages, reducing reproducibility barriers caused by model access restrictions or computational resource limitations. In addition, core modules, such as the screening mechanism for reward-exploration combinations and policy fusion logic, are implemented through modular design, with each component having independent functions and clear interfaces. Researchers can verify the functionality of each module individually, further enhancing the traceability of the reproducibility process.

A.4.3 DISCUSSION ON THE GENERALITY OF AFS

The generality of the Affordance State Space (AFS) in the PoRSE framework stems from the dual guarantee of "semantically driven design + dynamic adaptive mechanism," enabling it to function stably across different types of robotic tasks without extensive customization. From the perspective of interaction logic, AFS construction adheres to standardized prompt template specifications: when interacting with LLMs, only task goal descriptions and environmental state variable codes need to be replaced, while core components such as semantic mapping rules and dimension selection logic remain fixed. No task-specific prompt optimization is required, avoiding repeated development costs caused by task differences. Its built-in dynamic elimination mechanism further enhances task adaptability: this mechanism continuously tracks the actual contribution of each AFS dimension to policy performance, automatically selecting the exploration directions most aligned with task requirements in the current training phase. Even for unseen new tasks, such as collaborative multi-object sorting, or high-complexity tasks, such as dexterous hand grasping of irregularly shaped objects, it can quickly identify key state dimensions without manual intervention in dimension selection. Meanwhile, AFS exhibits strong fault tolerance toward invalid dimensions generated by LLMs: when LLMs propose task-irrelevant dimensions due to semantic understanding biases, the framework dynamically reduces the weight of such dimensions through policy performance feedback, and combines adaptive adjustment of the exploration reward coefficient $\beta$ to minimize the interference of ineffective exploration on training. Experiments show that even when using randomly assembled AFS dimensions—this setup is to simulate extreme noise scenarios—the framework's performance on typical tasks such as BlockStack and PushBlock remains significantly superior to that of traditional baseline methods. Furthermore, the consistent performance of AFS across different task types, including dexterous manipulation tasks like LiftUnderarm and legged locomotion tasks like Anymal walking, confirms that it can not only adapt to differences in task semantics but also be compatible with the state space characteristics of different robotic systems, providing core support for the cross-task expansion of the framework.

A.4.4 DISCUSSION ON EXPANDING PoRSE.

The method extensions of the PoRSE framework focus on addressing the adaptability challenges of complex real-world scenarios and further enhancing the framework's scalability, efficiency, and

stability, with several promising directions worth exploring. First, expanding the Affordance State Space (AFS) to multi-modal state representations is a key priority—by integrating multi-source sensory data such as RGB images, depth information, and tactile feedback, and combining vision-language models, such as CLIP or Grounding DINO, which are referred to as VLMs for short, the framework can automatically extract task-relevant affordance features from unstructured environmental inputs. For example, in cluttered domestic scenes with object occlusion, multi-modal AFS can fuse visual semantics, which involves identifying "mug" vs. "book", and tactile cues, which entail distinguishing "hard" vs. "soft" surfaces, to avoid over-reliance on pre-defined environmental semantics, enabling more robust exploration for dexterous manipulation tasks. Second, extending the in-policy-improvement grounding process—referred to as IPG—to hierarchical reinforcement learning architectures, or HRL for short, can effectively tackle long-horizon tasks that require sequential subtask decomposition, such as the "fetching ingredients, cutting, and cooking" process for a kitchen robot. By embedding IPG's dynamic reward-exploration optimization into each subtask layer, the framework can adjust the trade-off between global goal alignment and local subtask exploration—for instance, prioritizing exploration in the "cutting" subtask to find optimal knife angles while maintaining goal orientation in the "fetching" subtask, thereby improving the scalability of complex sequential tasks. Third, enhancing the elimination-expansion mechanism with meta-learning strategies can reduce computational overhead: by leveraging experience from historical tasks, the framework can meta-learn the optimal mutation rate adaptation rule—for simple single-object grasping tasks, a low mutation rate is used to stabilize policy optimization, while for complex multi-object collaborative tasks, a higher mutation rate is adopted to explore diverse reward-bonus combinations, minimizing trial-and-error costs. Finally, integrating uncertainty quantification into LLM-generated reward functions can mitigate instability in policy updates: by introducing Bayesian LLMs to estimate the confidence of reward rules—a typical case is that LLM-generated rewards in novel tasks with ambiguous semantics have lower confidence. Based on this, the framework can dynamically adjust the weight of exploration bonuses—when reward uncertainty is high, it increases exploration to collect more state feedback for reward refinement; when uncertainty is low, it focuses on goal-oriented optimization—avoiding policy oscillation caused by unreliable reward signals. These extension directions are mutually supportive: multi-modal state representations provide accurate state foundations for hierarchical RL, while meta-learning and uncertainty quantification jointly enhance the framework's efficiency and stability, ultimately enabling PoRSE to adapt to more complex real-world scenarios such as industrial assembly and home service robotics.

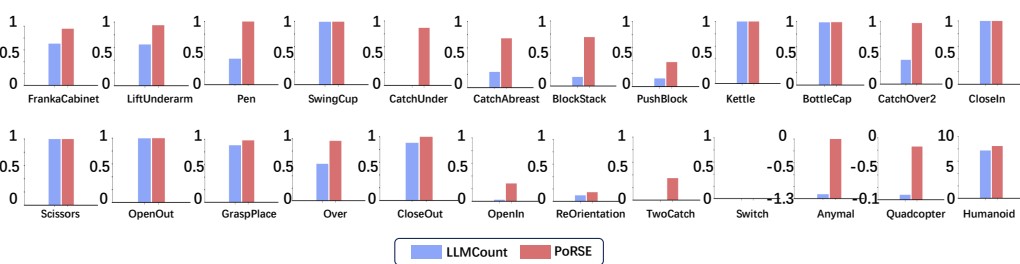

Figure 10: MTS Comparison of LLMCount and PoRSE methods, PoRSE method achieved higher MTS on 23 tasks.

### A.4.5 DISCUSSION ON BROADER IMPACT.

The development of PoRSE has broader implications for the field of robotics and AI. On the positive side, PoRSE's ability to efficiently acquire complex robotic skills using LLM-generated reward functions and directed exploration can accelerate the deployment of robots in various industries, including manufacturing, healthcare, and logistics. This can lead to increased automation, improved productivity, and enhanced safety in work environments. Moreover, the reduced need for manual reward design makes robotic skill learning more accessible to researchers and practitioners without extensive domain knowledge. However, there are also potential negative impacts and ethical considerations. The reliance on LLMs raises concerns about the transparency and interpretability of the

reward functions and exploration strategies. There is a risk of inheriting biases present in the LLM training data, which could lead to unfair or unsafe robotic behaviors.

## A.5   LIMITATION

Similar to other works utilizing large language models (LLMs) for content generation, our approach is also affected by the inherent instability commonly seen in LLM outputs. Even with identical prompts, the quality of generated functions can vary significantly, and in some cases, non-executable code may be produced. This instability is a prevalent limitation of LLMs rather than an issue unique to our approach, which can lead to a situation where, in each generation iteration (N=6 functions), only a part of functions may be successfully usable for guiding policy optimization. However, with the rapid development and iterative updating of LLM technology, it is anticipated that the instability in reward design and code generation will be progressively mitigated, and the capabilities of LLMs in these aspects will be steadily enhanced. By adopting the latest LLMs, frameworks like ours have the potential to achieve better robot skill learning outcomes.

To address this common limitation and ensure the fairness of comparison results in this paper, we have implemented the following measures for the Eureka, ROSKA, LLMCount, and PoRSE methods. For each iteration of functions generated by the LLM, we conduct code running correctness tests to ensure that all six functions generated in each iteration of the above methods can run correctly. This helps avoid result deviations caused by the instability of LLM responses and ensures that the comparison results are fair and reliable.

## A.6   PROMPT DETAIL

**Reward Function Initial Prompt Example**
You are a reward engineer trying to write reward functions to solve reinforcement learning tasks as effective as possible. Your goal is to write a reward function for the environment that will help the agent learn the task described in text. Your reward function should use useful variables from the environment as inputs. As an example, the reward function signature can be: {task_reward_signature_string} Since the reward function will be decorated with @torch.jit.script, please make sure that the code is compatible with TorchScript (e.g., use torch tensor instead of numpy array). Make sure any new tensor or variable you introduce is on the same device as the input tensors.

The output of the reward function should consist of two items:
(1) the total reward,
(2) a dictionary of each individual reward component.
The code output should be formatted as a python code string: ""'python ... ""'.

Some helpful tips for writing the reward function code:
(1) You may find it helpful to normalize the reward to a fixed range by applying transformations like torch.exp to the overall reward or its components
(2) If you choose to transform a reward component, then you must also introduce a temperature parameter inside the transformation function; this parameter must be a named variable in the reward function and it must not be an input variable. Each transformed reward component should have its own temperature variable
(3) Make sure the type of each input variable is correctly specified; a float input variable should not be specified as torch.Tensor
(4) Most importantly, the reward code's input variables must contain only attributes of the provided environment class definition (namely, variables that have prefix self.). Under no circumstance can you introduce new input variables.

**Reward Function Refinement Prompt Example**
Please carefully analyze the policy feedback and provide a new, improved reward function that can better solve the task. Some helpful tips for analyzing the policy feedback:
(1) If the success rates are always near zero, then you must rewrite the entire reward function
(2) If the values for a certain reward component are near identical throughout, then this means RL is not able to optimize this component as it is written. You may consider

(a) Changing its scale or the value of its temperature parameter
(b) Re-writing the reward component
(c) Discarding the reward component
(3) If some reward components' magnitude is significantly larger, then you must re-scale its value to a proper range
Please analyze each existing reward component in the suggested manner above first, and then write the reward function code.

**Mapping Function Initial Prompt Example**
You are a reinforcement learning engineer trying to write mapping functions to solve reinforcement learning tasks as effectively as possible. Your goal is to identify the variables for the environment that are maximally relevant in the task described in text. Some tasks may have only a single stage, and some tasks may have two separate stages.

You will be provided with a definition of the observation space for a reinforcement learning environment, and also provided with a small set of helper functions that can be used to transform the variables in the observation space. Write a function that returns the variable most associated with task for each stage of the task.

This function can take as input any member of self defined in compute_observations, and can apply any of the helper functions to any variables from self.obs_buf to generate new derived features (ex. computing the distance between object and goal). If a single stage requires multiple mapping variables, average the variables.

Also return a bool for each variable that is True if task goal requires the variable to increase, and False if it requires the variable to decrease.

Mapping_vars_name stores the string names of the corresponding variables in order.

**Mapping Function Refinement Prompt Example**
Please carefully analyze the policy feedback and provide an improved mapping function to better solve the task. Here are some tips for analyzing the policy feedback:

If the success rates are consistently near zero, you must rewrite the entire mapping function.

If the values of a mapping variable remain nearly identical throughout training, it means that the variable failed to effectively capture the task progress dynamics of the agent

You may consider:

(a) If one of the mapping variables in the current mapping function changes weakly during training and may not effectively quantify task progress, you should analyze its sensitivity and focus on whether the variable is strongly correlated with the task goal?
(b) If one of the mapping variables has not changed at all during the training process, indicating that the mapping variable cannot reflect the progress of the task. Please redesign a more sensitive alternative variable. requirement:
1. Strong correlation with task objectives
2. It can be decomposed into multi-stage indicators
(c) If one of the mapping variables does not significantly contribute to task performance, please analyze its necessity. If there is redundancy, please propose alternative mapping variables or directly eliminate the mapping variable
First, analyze each existing mapping variable using the guidelines above, then write an optimized mapping function.

**Policy Fusion $\alpha$ Search Prompt**
You are an expert in reinforcement learning, skilled in analyzing training data for policy models.

As we all know, in reinforcement learning, the policy continuously learns under the guidance of rewards to eventually achieve the task objective.

Currently, we train a policy over multiple stages. In each stage, corresponding rewards are used to guide the learning process. However, directly using the parameters of a policy trained

with the rewards of the previous stage as the initial parameters for the next stage often leads to learning issues due to changes in the rewards.

To mitigate this problem, we use a policy fusion method. Specifically, we perform a weighted sum of the parameters of the entire policy model and a random model:

$$\text{fusion} = \alpha \cdot \text{policy} + (1 - \alpha) \cdot \text{random}.$$

The current problem is determining the fusion ratio $\alpha$. I will provide you with the training data of the policy model from the previous stage. Please analyze the training results to determine the appropriate fusion ratio for the policy model.

Your task is to effectively analyze the training data of the current policy model. Here are some analysis techniques:

1. If the policy scores from the previous stage are consistently close to 0, it indicates that the rewards in the previous stage were ineffective in guiding the policy to achieve the reinforcement learning task objective. In this case, the fusion ratio $\alpha$ should be set lower to ensure higher plasticity in the policy model, allowing it to better adapt to the rewards in the next stage.

2. If the policy scores from the previous stage consistently increase, it indicates that the policy model was well-guided by the rewards to achieve the task objective. In this case, the fusion ratio $\alpha$ should be set higher to ensure the policy retains more effective experience.

3. If the policy scores from the previous stage remain constant but are not zero, it indicates that the rewards did not improve the existing policy. In this case, the fusion ratio $\alpha$ should be slightly reduced to increase the model's plasticity.

Since a single fusion ratio may not be completely accurate, you will have five opportunities to design this parameter. In these five attempts, you should:

1. Propose the most likely optimal fusion ratio $\alpha$.
2. Ensure the ratios $\alpha$ are distributed as evenly as possible in the range from 0 to 1, avoiding clustering all five fusion ratios within a narrow range.

**reward balance $\beta$ Search Prompt**
You are an expert in reinforcement learning, skilled in analyzing training data for policy models. As we all know, the training of a policy model requires effective guidance from rewards. At different stages of policy model training, adjustments should be made based on the characteristics of different tasks.

Currently, we have two types of rewards jointly guiding the training of the policy model, so the coefficient ratio between these two rewards is crucial.

The first reward is the dense reward designed by the large model (LLM_Reward_Ratio). This reward directly comes from the reward function designed by the large model, and its primary focus is always on achieving the task objective to guide policy training. However, whether the reward function designed by the large model can effectively guide the policy training is uncertain. But the goal of this reward is known: to always guide the policy to achieve the corresponding reinforcement learning task objectives.

The second reward is the progress-based exploration reward (Progress_Reward_Ratio). By utilizing the progress function designed by the large model for different reinforcement learning tasks, the progress function converts high-dimensional environmental state information into low-dimensional progress variable information related to task completion.

This progress variable measures the task completion progress. By dividing the continuous progress variable into multiple numerical intervals, exploration rewards are based on the number of visits to each interval. That is, the fewer visits to a certain progress interval, the higher the exploration reward. Conversely, the more visits, the lower the reward.

This encourages the policy model to continuously advance the task completion progress.

Now, I will provide you with the training data of the current policy model over a recent period. This training data includes each component of the reward function designed by the large model, each progress variable of the progress function, and the current task scores.

These data are presented in array format. During this training period, your task is to analyze the training data effectively. Here are some analysis techniques:

1. If the current reinforcement learning task scores remain nearly the same, it indicates that the reward function designed by the large model cannot effectively guide the policy to achieve the task objective. Therefore, the coefficient of the LLM_Reward component should be slightly reduced.

2. If the reinforcement learning task scores increase gradually during the training process, it indicates that the reward function designed by the large model, combined with the exploration reward, can effectively guide the policy to achieve the task objective. In this case, the coefficient of the LLM_Reward component can be slightly increased.

Please determine the optimal ratio of these two reward components based on the information provided. The coefficient range for both rewards should be controlled between 0 and 2. Since a single attempt may not be precise, you will have five opportunities to design parameters.

In these five attempts, you must not only propose the most likely optimal reward parameter ratios but also maintain a relatively uniform search distribution, avoiding concentrating all five designs in the same parameter interval.

