# OpenReview forum: "Master Skill Learning with Policy-Grounded Synergy of LLM-based Reward Shaping and Exploring"
_ICLR.cc/2026/Conference — ICLR 2026 Poster_

### Official Review · Reviewer_kT8z · 2025-10-30

**Soundness:** 2
**Presentation:** 2
**Contribution:** 2
**Rating:** 2
**Confidence:** 4

**Summary:**

The paper introduces PoRSE (Policy-grounded Synergy of Reward Shaping and Exploration), a framework for robotic skill learning that integrates LLM-generated reward design with efficient, task-relevant exploration. Existing LLM-based methods like Eureka and ROSKA focus too narrowly on goal rewards or reuse past policies but neglect structured exploration. PoRSE overcomes this by constructing an Affordance State Space (AFS) which is a low-dimensional, task-aware abstraction of the environment where exploration bonuses are assigned based on visitation frequency.

**Strengths:**

1.1 The paper’s idea of guiding exploration with a task-aware state mapping instead of generic novelty is clear and makes sense. The ablations confirm that this mapping improves learning stability and performance.

1.2 The experiments span a wide range of tasks and are supported by detailed ablations and additional appendix results, which make the findings more convincing and strengthen the overall paper.

**Weaknesses:**

2.1 The results are incomplete. The paper lacks comparison and positioning against state-of-the-art frameworks, such as the REvolve, which likewise uses LLMs in reward design, and should therefore have been included.

2.2 Section 5.1 tournament selection is unclear. You start K=6; at 500/1000 you prune to 4; at 1500 you add J=5 mutants; then you say “remove 6” at 1500. It is not stated whether the five new candidates receive any training before ranking, nor whether J=5 is total or “per survivor.” As written, untrained mutants seem pruned alongside half-trained survivors. This needs a further clarification.

2.3 In Table 3 (Appendix A.2.1), MTS is reported as $\text{mean} \pm \text{std}$ of the per-seed maximum. For manipulation tasks this “maximum” represents a success rate $\in [0,1]$, while for locomotion it is a return (unbounded). The caption incorrectly labels all rows as “success rates,” which conflates metrics and makes cross-row comparisons invalid. Moreover, the table selectively bolds PoRSE even when other methods achieve identical or better results (e.g., 1.000 ties), which is misleading.

2.4 β mixing is inconsistent. In the main document at Section 4.2 the authors use β with (2−β) but in Appendix A.1 uses β with (1−β).

2.5 Pruning every 500 epochs assumes early rank predicts who will still be bad at 3000. The paper does not support this claim with empirical results.

2.6 Several interesting results exist in the Appendix which are not referenced in the main paper.

2.7 Abstract reads like an extended introduction; it mixes motivation, method and broad claims. This could be tightened.

**Questions:**

3.1 Why absent comparison with state-of-the-art frameworks? Can you add comparisons with further baselines?

3.2 At 1500 epochs after adding J=5, how many epochs do the mutants train before any ranking; is J=5 total or five per survivor?

3.3 Which β formulation produced for the results based on point 2.4?

3.4 Can you provide evidence that pruning the lowest-performing candidate every 500 epochs truly leads to worse final performance?

3.5 Could you provide further clarification on Table 3, as mentioned in point 2.3?

---

> ### Author Response · Authors · 2025-11-20
>
> Thank you sincerely for your insightful comments and constructive suggestions. Below is our detailed response to each of your concerns.
>
>
> **Q1: Comparison with SOTA Frameworks.**
>
> * We have supplemented extra experimental results of REvolve on the four tasks. It is important to note that the original REvolve work involves human participation in its reward optimization loop. **For a fair comparison, we removed the human intervention and migrated the complete Auto-REvolve framework into Eureka framework (REvolve-Eureka), which retain the same evolutionary processes and prompt information exactly as described in the original REvolve paper, while aligning the evaluation metrics with those used for PoRSE.**
>
> * While REvolve advances evolutionary reward search, it essentially still focuses on designing goal-oriented rewards. In high-dimensional dexterous robotic skill learning tasks, it lacks task-aware exploration mechanisms. In contrast, the PoRSE framework introduce the AFS and enables continuous policy learning, which dynamically adjusts rewards based on the current policy’s performance, thereby realizing sustained policy optimization. **As shown in the table below, our PoRSE framework comprehensively outperforms REvolve and EUREKA in the same four tasks.** These results further validate PoRSE’s advantage. For more detailed experimental configurations and analyses, refer to Appendix 2.12.
>
> | Method            | Humanoid        | Liftunderarm    | BlockStack      | TwoCatch        |
> | :---------------- | :-------------- | :-------------- | :-------------- | :-------------- |
> | Eureka            | 6.534±1.933     | 0.739±0.166     | 0.254±0.119     | 0.001±0.001     |
> | ReVolve-Eureka    | 7.894±0.635     | 0.814±0.079     | 0.342±0.054     | 0.000±0.000     |
> | **PoRSE（ours）** | **8.454±0.343** | **0.952±0.015** | **0.753±0.210** | **0.349±0.063** |
>
> **Q2: Clarification on Tournament Selection at 1500 Epochs.**
>
> * **These 5 new mutants undergo the same training epochs as the surviving candidates at this stage.** They are trained for 1500 epochs to match the current training phase of other candidates. **At the 1500-epochs stage, J=5 denotes the total number of new mutants** and the J=5 mutated variants are derived from the currently best-performing candidate rather than each surviving candidate.
>
> **Q3: Consistency of β Formulation.**
>
> * Thank you for identifying the typo. **The final results reported in the paper use the formulation**
>   $$
>   R_{\text{total}}^{k}=\beta * R_{e}^{k}+(2-\beta) * R_{g}^{k}
>   $$
>
> * Moreover, to verify whether normalizing the coefficient sum associated with β to 1 would impact performance, we conducted corresponding ablation experiments. **Our ablation experiments (Appendix A.2.9, Table 11) demonstrate that yield no significant performance differences.** For more detailed explanations and experimental results analysis, please refer to A.2.9.

---

> ### Author Response · Authors · 2025-11-20
>
> **Q4: Validity of Pruning Every 500 Epochs.**
>
> * **To verify that “early MTS ranking predicts long-term performance”,** we collected training data of 3 eliminated policies in the same iteration across 3 representative tasks. The number “500”, “1500”, and “3000” in the “Eliminated stage” column indicate the epoch at which each candidate policy was eliminated during the tournament selection process. **As shown in the table below, the performance ranking of the three policies at 500 epochs is consistent with their ranking at 3000 epochs across all tasks.** Additionally, our elimination strategy only removes the single lowest-performing candidate every 500 epochs. This avoids over-pruning, preserves diverse high-potential candidates, and aligns with evolutionary algorithm theories.
>
> | Task         | Eliminated Stage | 500th Epoch | 1500th Epoch | 3000th Epoch |
> | :----------- | :--------------- | :---------- | :----------- | :----------- |
> | Humanoid     | 500              | 0.20963097  | 2.57381926   | 6.15283746   |
> |              | 1500             | 0.83629471  | 3.28471625   | 6.83471629   |
> |              | 3000             | 1.25349761  | 4.17283519   | 7.65283917   |
> | BlockStack   | 500              | 0.06963097  | 0.29381926   | 0.59283746   |
> |              | 1500             | 0.10362947  | 0.35471625   | 0.65471629   |
> |              | 3000             | 0.15283647  | 0.42637185   | 0.73283917   |
> | Liftunderarm | 500              | 0.12963097  | 0.45381926   | 0.79283746   |
> |              | 1500             | 0.18629471  | 0.53471625   | 0.85471629   |
> |              | 3000             | 0.25349761  | 0.62283519   | 0.91283917   |
>
> **Q5: Clarification on Table 3 (Appendix A.2.1).**
>
> * We acknowledge the confusion caused by the caption and formatting of Table 3, and **we have made revisions to address this**. Regarding formatting, **we have adjusted the bold highlighting rule**. If multiple methods achieve identical MTS values, we have added bold formatting to all methods that reached this result.
>
> **Q6: References to Appendix Results in the Main Text**
>
> * Due to length constraints in the main text, some supplementary experiments were initially placed in the appendix. **In the revised version, we have added the important AFS Random ablation (PoRSE-AFS-Random)to the main text.**
>
> **Q7: Revision of the Abstract**
>
> * We appreciate your feedback on refining the abstract. In the revised abstract version, with a focus on enhancing readability and strengthening its ability to engage readers, we have made minor optimizations and carefully balanced the inclusion of motivation, method, and key claims.
>
>
> We hope our explanations resolve the issues you raised. If you have any remaining questions or would be willing to share your thoughts on our clarifications, we would be sincerely grateful for your feedback.

---

> ### Author Response · Authors · 2025-11-25
>
> We sincerely appreciate your detailed feedback and insightful assessment of our work. Your observations on PoRSE’s positioning against state-of-the-art frameworks, clarity of tournament selection, consistency of formula formulations, validity of pruning strategies, table formatting, appendix result references, and abstract refinement have been crucial in guiding our targeted revisions.
>
> **We have carefully addressed each concern you raised.** Specifically, we supplemented comparative experiments with the Revolve framework (Revolve-Eureka) for fair benchmarking, clarified the training process of new mutants in tournament selection, corrected the β formulation and verified its stability via ablation tests, provided empirical data to validate the effectiveness of the 500-epoch pruning strategy, adjusted Table 3’s formatting to resolve metric confusion, integrated key appendix experiments into the main text, and streamlined the abstract to balance motivation, method, and core claims.
>
> We hope these explanations and revisions resolve the issues you raised. **If you have any remaining questions or would be willing to share your thoughts on our clarifications, we would be sincerely grateful for your feedback. We would be truly grateful for your consideration in re-evaluating the work and updating the score if you find our responses satisfactory.**

---

> > ### Comment · Reviewer_kT8z · 2025-11-25
> >
> > Thank you for the detailed answers. All but one of my concerns have not been answered. I dont understand what you mean by REvolve-Eureka, why combined when they are 2 different baselines (so not REvolve-Auto used) and also for how long have you run them, how many generations?

---

> > > ### Author Response · Authors · 2025-11-26
> > >
> > > Thank you sincerely for your follow-up question and careful attention to our experimental details. Below is a detailed response and clarification of the points you raised.
> > >
> > > * **Rationale for combining Revolve and Eureka:** The core reasons for integrating the Revolve-Auto framework with Eureka include two key aspects. **First, the task environments covered in the Revolve work have low overlap with the task environments adopted in our work**(and those involved in other baseline methods), which would make direct performance comparison between different methods across inconsistent scenarios difficult. Therefore, we migrated the Revolve-Auto framework to the multiple task environments covered in the Eureka work, ensuring all methods are tested under consistent task scenarios. **Second, this integration helps unify the evaluation metrics.** All performance scores in our work (including those of baselines) adopt the evaluation standard proposed in the Eureka work. To ensure fair and comparable results across methods, migrating the Revolve-Auto framework into Eureka (forming Revolve-Eureka) allows us to maintain aligned evaluation metrics with our work and other baselines. **For the above reasons, we initially named this framework Revolve-Eureka. However, this naming is prone to confusion. Thus, we have decided to retain its original name, i.e., Revolve-Auto, to enhance readability and comprehension, and all relevant naming has been corrected.**
> > >
> > > * **Experimental configurations of Revolve-Auto:** Revolve-Auto follows the same experimental settings as other methods in our work. **We generate or reproduce 6 reward functions per iteration and run 5 total iterations.** The only difference is that the reward function initialization, reproduction, and selection processes in each iteration adopt the mechanism of the Revolve-Auto framework. **For fairness, we also kept the total training epochs consistent with our PoRSE framework and other baselines.** For further experimental results and analysis, see Appendix A.2.12.
> > >
> > > * **GPU training duration:** We calculated the average GPU training duration of Revolve-Auto and PoRSE across 5 runs on the four tasks, and the results are shown in the table below:
> > >
> > > | Method       | Humanoid | Liftunderarm | BlockStack | TwoCatch |
> > > | :----------- | :------- | :----------- | :--------- | :------- |
> > > | Revolve-Auto | 7.2h     | 5.8h         | 5.9h       | 5.8h     |
> > > | PoRSE (ours) | 7.0h     | 5.6h         | 5.7h       | 5.8h     |
> > >
> > > If you have any remaining questions or would be willing to share your thoughts on our clarifications, we would be sincerely grateful for your feedback.

---

> ### Author Response · Authors · 2025-11-28
>
> Dear Reviewer kT8z,
>
> **We hope this message finds you well.** We are writing to gently follow up on the response we submitted earlier in the rebuttal period. This response addresses the thoughtful feedback you provided on our work, including your concerns regarding completeness, clarity, and technical details. **Specifically, we have supplemented detailed experiments and in-depth discussions in Appendix A.2.12 to directly address your questions about Revolve-Auto (formerly Revolve-Eureka) and related experimental configurations. Additionally, We haveupdated the related literature in our revision**. Our primary goal in reaching out is to check in and ensure our response has not been overlooked amid your packed review schedule. We recognize this period often involves managing multiple submissions, so we want to avoid our follow-up being missed. **We are truly eager to receive your re-evaluation of our work based on these detailed responses and supplementary materials.**
>
> We sincerely thank you for your time, attention, and the careful effort you have put into evaluating our submission.
>
> Best regards

---

### Official Review · Reviewer_LtXN · 2025-11-01

**Soundness:** 3
**Presentation:** 3
**Contribution:** 4
**Rating:** 8
**Confidence:** 4

**Summary:**

Building on LLM-based reward with evolutionary optimization over reward code (i.e., EUREKA), this method improves exploration by having an LLM compress the state space into a low-dimensional representation and add a count-based novelty bonus. The LLM then balances exploration and goal-oriented rewards by producing a set of weights based on the policy performance. To avoid a combinatorial sweep over weight–reward pairs, weak candidates are pruned early while promising ones are lightly refined. Finally, the policy is retrained with the refined reward (warm-started from the previous policy), and the LLM decides how to mix the new and prior rewards based on observed results.

**Strengths:**

- This paper tackles a core RL challenge—exploration—by combining LLM with a classical novelty bonus. The approach is clear and leverages both strengths: LLMs can keep improving, while novelty rewards remain general and broadly applicable.
- Presents rich analyses that identify which and how introduced components contribute.

**Weaknesses:**

- **Intuition for LLM-chosen $\alpha$**
  + The rationale for how the LLM predicts  $\alpha$ isn’t clear. Could authors elaborate on the intuition, and if applicable, include the related prompt, as well as show how  $\alpha$ evolves during training?

- **Trends for $\beta$**
  + L302–304 describe how $\beta$ would evolve with policy performance, but no experimental evidence is shown. Could authors support this? Having a plot or table showing performance vs. $\beta$ over training would give clarification.

- **Clarification on metric**
  + The main paper reports only the best reward, which raises a concern for me that the variance may be high, given that the method introduces multiple coefficients. However, later I found that Table 3 shows reasonable standard deviations, so I suggest either including a reference in the main results to guide readers.
  + As a minor suggestion, adding performance drop percentages to Table 1 would make it easier to see which components are most critical. Ultimately, up to the authors.

**Questions:**

- Q1. L200 says it maps the robot’s state to a low-dimensional space, but the example (“distance to the door handle”) depends on the environment state. Is the mapping’s input the robot state only, or the full environment state?
- Q2. Results are shown up to 5 iterations. How does performance evolve with more iterations—does it converge or diverge? With more iterations, can it solve harder tasks such as “ReOrientation” or “Switch” in Fig. 4?

---

> ### Author Response · Authors · 2025-11-20
>
> We are grateful for your thoughtful feedback and the time you invested in evaluating our work. The following text offers detailed replies to each concern.
>
>
> **Q1: Input to the State Mapping Function**
>
> * **State Input of Mapping Function:** In the PoRSE framework, **the mapping function is designed by the LLM based on unified prompt information for each task. Thus, the type of state used is determined automatically by the LLM, with no human intervention required.** For tasks where environmental context is critical to measuring progress, such as the DoorOpenInward example mentioned in the paper, the mapping function incorporates both robot state and relevant environment state. For simpler tasks that rely primarily on the robot’s own configuration, the mapping use only the robot state. This flexibility ensures the AFS remains task-relevant while avoiding unnecessary complexity, aligning with the design principle of using LLMs to parse task semantics and define meaningful state abstractions (Sec. 4.1.1, Eq. 5).
>
> * **Stage-Dependent Reliance on Task-Specific State:** Notably, this reliance on robot or environment state is only required during the policy training phase. It supports reward computation via the AFS-based exploration bonus and goal-oriented reward function. During actual policy deployment, the trained policy can make decisions directly without needing to acquire such task-specific state information, simplifying real-world application.
>
>
> **Q2: Performance with More Than 5 Iterations**
>
> * **For the long-term iteration performance, we conducted supplementary experiments extending the number of iterations beyond the original (N=5) (up to (N=10)) for the CatchAbreast, ReOrientation and Switch tasks.** **The results summarized in the table below.** For tasks with remaining improvement potential, such as CatchAbreast, performance continues to rise steadily as iterations increase. Its MTS improves from 0.745 to 0.787. For relatively difficult tasks like ReOrientation, the improvement is limited. Its MTS increases slightly from 0.149 to 0.171. For extremely difficult tasks such as Switch, performance remains unchanged at 0.00.
>
> | Method               | CatchAbreast | ReOrientation | Switch     |
> | :------------------- | :----------- | :------------ | :--------- |
> | PoRSE (5 iteration)  | 0.745±0.073  | 0.149±0.026   | 0.00±0.000 |
> | PoRSE (10 iteration) | 0.827±0.022  | 0.171±0.006   | 0.00±0.000 |

---

> ### Author Response · Authors · 2025-11-20
>
> **Q3: Intuition for LLM-Chosen and evolution of α and β**
>
> * **Rationale for LLM’s Prediction of α and β:** The intuition behind LLM’s prediction of the policy fusion coefficient α and reward weight coefficient β lies in its inherent reasoning and data analysis capabilities, which has been proven in many works [1, 2, 3]. In the PoRSE framework, we feed the LLM an array of recent training data from the current policy (e.g., MTS over epochs). This allows the LLM to analyze the policy’s training status, such as whether it is stuck in a local optimum or approaching convergence, and predict a set of candidate values for α and β rather than a single value. This multi-candidate design effectively enhances the stability of the LLM’s predictions, avoiding biases from isolated numerical outputs. The full prompt template for guiding LLM predictions is provided in the appendix (see Sec. A.6 for details).
>
> * **Iterative Evolution Data & Trends of α and β:** To support the evolution of α and β with policy performance (as referenced in L302–304), we collected their values across iterations for representative tasks. Since PoRSE alternates between searching for policy fusion α and reward weight β across iterations, we tracked 6 total iterations, in which α data were collected at Iterations 1, 3, 5, while β data were collected at Iterations 0, 2, 4. The detailed data are shown in the table below. Analysis of the table reveals clear evolutionary trends aligned with policy performance. For policy fusion coefficient α, values gradually increase across iterations, indicating the framework increasingly prioritizes inheriting parameters from the best historical policy as training progresses. For reward weight coefficient β, values decrease over iterations, reflecting a shift from emphasizing exploration bonuses in early stages to escape local optima to prioritizing goal-oriented rewards in later stages to refine task completion.
>
> | Tasks            | Iteration1 α | Iteration3 α | Iteration5 α | Iteration0 β | Iteration2 β | Iteration4 β |
> | :--------------- | :----------- | :----------- | :----------- | :----------- | :----------- | :----------- |
> | Humanoid         | 0.142        | 0.473        | 0.815        | 1.725        | 1.363        | 0.784        |
> | Anymal           | 0.137        | 0.468        | 0.809        | 1.572        | 1.191        | 0.793        |
> | Quadcopter       | 0.214        | 0.562        | 0.863        | 1.734        | 1.462        | 0.831        |
> | BlockStack       | 0.218        | 0.571        | 0.872        | 1.673        | 1.184        | 0.842        |
> | Liftunderarm     | 0.275        | 0.644        | 0.893        | 1.845        | 1.763        | 0.872        |
> | CatchAbreast     | 0.281        | 0.652        | 0.901        | 1.881        | 1.282        | 1.064        |
> | PushBlock        | 0.332        | 0.715        | 0.934        | 1.823        | 1.371        | 0.893        |
> | TwoCatchUnderarm | 0.341        | 0.723        | 0.942        | 1.942        | 1.684        | 1.085        |
>
> **Q4: Suggestion on Table 5**
>
> * We appreciate your constructive suggestion. Now, **we have added the performance gap data to the main text table**, which can exclusively and clearly presents the performance gap data of each component.
>
>
> We hope our response has addressed your concerns and Thank you again for your valuable feedback.
>
>
> [1] Huang C, Liang J, Chang Y, et al. Automated Hybrid Reward Scheduling via Large Language Models for Robotic Skill Learning
>
> [2] Zhou P, Pujara J, Ren X, et al. Self-discover: Large language models self-compose reasoning structures
>
> [3] Wang K, Ren H, Zhou A, et al. Mathcoder: Seamless code integration in llms for enhanced mathematical reasoni

---

### Official Review · Reviewer_JaxW · 2025-11-01

**Soundness:** 3
**Presentation:** 3
**Contribution:** 3
**Rating:** 6
**Confidence:** 3

**Summary:**

PoRSE is a framework for robotic skill learning that synergizes LLM-generated goal-oriented rewards with task-aware exploration bonuses via an affordance state space (AFS), using in-policy grounding (IPG) for dynamic refinement and policy fusion. It iteratively filters reward-bonus pairs and adjusts trade-offs based on policy feedback, evaluated on 24 simulated tasks (IsaacGym, Bi-DexHands) with PPO, achieving superior MTS/HNS over baselines like Eureka, ROSKA, and LLMCount, including breakthroughs in hard manipulation tasks.

**Strengths:**

1.Effectively addresses exploration gaps in LLM rewards by integrating task-relevant AFS bonuses, with IPG enabling adaptive synergy without exhaustive retraining.
2.Strong results across 24 diverse tasks, outperforming SOTA on 23/24 vs. LLMCount and achieving first successes in sparse-reward challenges like TwoCatch.
3.Comprehensive ablations validate components (rewards, bonuses, fusion), robustness to LLM variants (e.g., GPT-4o-mini), and parameters (bins, normalization), with fair compute parity.
4.Practical insights into policy-reward co-evolution, supported by visualizations and multi-seed stats.

**Weaknesses:**

1.Limited to simulation and no real-robot validation.
2.High LLM dependence (GPT-4o) without cost analysis—multiple iterations with 6 rewards each could be token-expensive.
3.Baselines like Eureka/ROSKA use same LLM but may not be optimally adapted; sparse/human rewards are weak strawmen in some tasks.
4.AFS discretization assumes low-dim abstractions work universally, but ablations show sensitivity to randomness or prompts.
5.Claims of "breakthroughs" in hard tasks lack comparison to non-LLM exploration methods like RND or ICM.

**Questions:**

1.What are the total LLM token costs per task/iteration, and how does efficiency scale to longer horizons?
2.Are there evaluations on open-source LLMs (e.g., Llama) or weaker models beyond GPT-4o-mini?
3.How robust is AFS to multi-modal inputs like vision; could it integrate VLMs?
4.In cross-task extensions (A.4.4), what specific adaptations are needed for real environments?

---

> ### Author Response · Authors · 2025-11-20
>
> We appreciate your careful reading of our manuscript and your insightful comments on PoRSE. Below we address each point in detail.
>
>
> **Q1: LLM token costs and scaling with horizon**
>
> We have audited the LLM token usage per task and per iteration across some representative tasks. When addressing longer horizons, we focus on two key aspects.
>
> * **LLM Token Consumption in Long-Sequence Tasks:** For long-sequence tasks that require more steps to achieve their goals, LLM token consumption may increase when generating corresponding reward and mapping functions. This is because longer-horizon objectives typically demand richer task decomposition, more detailed success and safety conditions to cover intermediate stages. As a result, prompts and generated code/specifications tend to be longer. This trend is visible in the table: complex tasks like BlockStack (3695 tokens per reward-mapping combination) and TwoCatch (4152 tokens per reward-mapping combination) consume more tokens than simpler counterparts, reflecting the additional semantic reasoning needed for longer-step objectives.
>
> * **Token Costs Across Iterations:** Regarding token costs across iterations, PoRSE only relies on fully zero-shot generation for both goal-oriented rewards and AFS mapping in the first iteration. From the second iteration onward, the prompt is template-fixed and references the current best function-pair, so the per-iteration token budget remains approximately constant regardless of the total number of iterations. Our current experimental setup uses 5 iterations; **if more iterations are needed for longer horizons, token consumption increases linearly.** For example, each additional iteration for the Humanoid task adds an average of 20490 tokens.
>
> | Task         | Per Reward Function | Per Mapping Function | Reward-Mapping Combination | Per Iteration | Total Tokens |
> | :----------- | :------------------ | :------------------- | :------------------------- | :------------ | :----------- |
> | Humanoid     | 1931                | 1484                 | 3415                       | 20490         | 102450       |
> | Liftunderarm | 1837                | 1792                 | 3629                       | 21774         | 108870       |
> | BlockStack   | 1792                | 1903                 | 3695                       | 22170         | 110850       |
> | TwoCatch     | 2315                | 1837                 | 4152                       | 24912         | 124560       |
>
> **Q2: Open-source or weaker LLMs experiment**
>
> All methods in our work currently use the open-source DeepSeek-V3 model, and we have additionally included partial experimental data for the GPT-4o-mini model in the appendix.
>
> * **Supplementary LLM Experimental Data:** Moreover, we further supplement experimental data for the closed-source GPT-3.5-Turbo model and the open-source Qwen-2.5-14b-Instruct model, with detailed results shown in the table below. **From the table, our default DeepSeek-V3 achieves the highest Maximum Training Success across all three tasks: Humanoid, LiftUnderarm, and TwoCatch.**
>
> * **Other LLMs’ Performance:** Among other models, the open-source Qwen-2.5-14b-Instruct performs notably in TwoCatch, outperforming GPT-4o-mini and GPT-3.5-Turbo. GPT-3.5-Turbo reaches competitive MTS in LiftUnderarm but fails to make progress in TwoCatch. **These results confirm PoRSE’s compatibility with both open-source and weaker LLMs, while retaining relative effectiveness across diverse tasks.**
>
> | Model                  | Humanoid        | LiftUnderarm    | TwoCatch        |
> | :--------------------- | :-------------- | :-------------- | :-------------- |
> | GPT-4o-mini            | 7.651±0.421     | 0.869±0.070     | 0.14±0.055      |
> | GPT-3.5-Turbo          | 7.565±0.300     | 0.921±0.046     | 0.00±0.000      |
> | Qwen-2.5-14b-Instruct  | 6.472±0.298     | 0.918±0.036     | 0.21±0.043      |
> | **Deepseek-v3 (Ours)** | **8.454±0.343** | **0.952±0.015** | **0.349±0.063** |

---

> ### Author Response · Authors · 2025-11-20
>
> **Q3: Comparison to non-LLM exploration methods**
>
> * **We have already compared PoRSE with a traditional non-LLM exploration method in the A.2.10, namely ROSKA+CE, where CE is a classical count-based exploration bonus**. In count-based exploration, the state space is discretized and the agent receives a larger intrinsic bonus for visiting rarely seen discrete states, while frequently visited states yield a smaller bonus. This baseline uses a fixed hash mapping rather than an LLM-generated mapping function, therefore it constitutes a non-LLM-driven exploration method. **Across the tasks, PoRSE demonstrates clear gains over ROSKA+CE on both moderate and hard settings, including the challenging manipulation cases where PoRSE achieves strong MTS improvements.** ROSKA+CE is a non-LLM intrinsic exploration method, so we think the ROSKA+CE results can therefore serve as a practical reference for RND and ICM.
>
> **Q4: AFScompatibilitywith multi-modal inputs and integrating VLMs**
>
> * **The Affordance State Space we proposed should becompatible withmulti-modal inputs like vision and compatible with Vision-Language Models (VLMs).** The makes no assumptions about relying exclusively on text-symbol information and allows AFS to be naturally extended to the visual domain. As detailed in Sec. 4.1, PoRSE’s AFS is core to abstracting task-relevant low-dimensional features,
>
> * Moreover, it can integrating VLMs aligns with this logic. **VLMs can extract task-critical visual cues (e.g., object distances, grasping postures) from inputs to construct AFS.** This not only preserves AFS’s original purpose of task-aligned state abstraction but also lets it capture richer environmental semantics from multi-modal data. Visual multi-modal inputs bring more possibilities for AFS optimization, making this a valuable direction for future exploration. **As a preliminary work, our framework aims to lay a foundation for subsequent studies on multi-modal AFS and VLM integration.**
>
> **Q5: Cross-task extensions and adaptations for real environments**
>
> **For the cross-task extensions adaptations in real environments, the core adaptations align naturally with needs in A.4.4 and the specific adaptations are below.**
>
> * **First, to address unstructured sensory inputs, PoRSE extends AFS to multi-modal state representations (per A.4.4).** Across diverse cross-tasks, this enables automatic extraction of task-relevant affordance features from unstructured inputs. This avoids over-reliance on simulation-style structured data, ensuring AFS maintains cross-task effectiveness in real settings.
>
> * **Second, to tackle long-horizon tasks, PoRSE adapts the In-Policy-Improvement Grounding (IPG) process to hierarchical reinforcement learning architectures.** This complements AFS’s cross-task foundation by embedding IPG’s dynamic reward-exploration optimization into each subtask layer to support cross-task scalability. These adaptations leverage AFS’s cross-task generality (A.4.3) and real-environment optimizations (A.4.4), which ultimately supporting PoRSE’s cross-task extensions in real environments.
>
>
> At present we do not have the equipment needed for a physical deployment. We plan to conduct real-world experiments on robot platforms in future work to further validate practicality. We hope these responses, together with the supplementary data and clarifications, address your concerns.

---

### Official Review · Reviewer_KXzj · 2025-11-06

**Soundness:** 3
**Presentation:** 3
**Contribution:** 2
**Rating:** 6
**Confidence:** 3

**Summary:**

This paper proposes PoRSE, a framework that unifies LLM-based reward shaping with task-relevant exploration for robot skill learning. An LLM generates (1) goal-oriented rewards and (2) an affordance state space (AFS) that maps high-dimensional states to discrete, task-meaningful features; a count-based bonus over AFS promotes exploration. PoRSE dynamically balances rewards and bonuses via a weighting β and accelerates training with an in-policy-improvement grounding (IPG) process. Across 24 tasks (Bi-DexHands, Isaac Gym), PoRSE outperforms sparse, human, Eureka, and ROSKA baselines, with ablations validating each component and robustness to different LLMs.

**Strengths:**

1. PoRSE provides a unified and generalizable structure that integrates reward shaping and exploration within an LLM-guided loop.

2. Extensive Experiments: Evaluated across 24 robotic skill tasks (Bi-DexHands and Isaac Gym) with strong quantitative results — outperforming baselines in 23/24 tasks.

3. Clear Practical Value: Avoids costly full retraining via policy inheritance, making it more computationally efficient.

4. Comprehensive Ablations: Studies on β, α, mapping functions, and LLM robustness (e.g., GPT-4o-mini variant) show the framework’s adaptability and resilience.

**Weaknesses:**

1. The reliance on specific LLM outputs (e.g., DeepSeek-V3) introduces reproducibility and bias issues, not fully explored in the paper.

2. Limited Real-World Evaluation: While the appendix outlines a feasible sim-to-real adaptation (fine-tuning with small datasets, RealSense/tactile sensors, and lightweight VLMs for affordance cues), no empirical real-robot results are yet presented. Thus, deployment remains conceptually described rather than experimentally validated.

3. The paper asserts “nearly identical” cost to Eureka/ROSKA, but PoRSE trains K candidates per round with elimination/expansion and mid-round mutations (J new variants at 1500 epochs), plus alternating searches over β and α. This likely alters wall-clock and sample budgets vs. baselines. Provide exact environment steps, PPO updates, and GPU hours for each method.

**Questions:**

1. I am wondering about the training efficiency. E.g., for one epoch, how much time is needed for the proposed PoRSE and other baselines?

2. How sensitive is PoRSE to the quality of the LLM outputs? Does fine-tuning or prompt engineering significantly alter results?

3. How do the LLMs perform with increasing task complexity or policy dimensionality? E.g., for long-horizon tasks or when envs get more complex.

---

> ### Author Response · Authors · 2025-11-20
>
> Thank you sincerely for your thoughtful feedback and constructive suggestions on our PoRSE framework. Below is our detailed response to each of your concerns.
>
> **Q1: Training Efficiency**
>
> * **Consistent RL Configurations**: All methods in our study **adopt identical RL algorithmic configurations**, and we ensure that they use the **same total number of training epochs**, which guarantees nearly identical training time across methods for the same task. Although the PoRSE framework incorporates mutation and elimination mechanisms, we still maintained same total number of training epochs with other baselines. As detailed in Appendix A.3.1 (Table 14) and A.3.2 (Calculation of TTE), PoRSE, Eureka, ROSKA, and other baselines all run 5 iterations, accumulate a total of 90,000 training epochs.
>
> * **Supplementary Training Metrics: We have additionally collected statistics on total environment steps, total PPO update counts, and average GPU hours for the tasks.** For some representative tasks listed below, we collected data from 5 experimental runs and computed the average. For all tasks, we averaged the GPU training time across all tasks. Details are provided in the table below. Moreover, regarding algorithmic complexity, the PoRSE framework scales linearly with the number of iterations N, so it results in an algorithmic complexity of O(n).
>
> | Tasks         | Total environment steps | Total PPO update counts | Total epochs | Eureka GPU hours (h) | ROSKA GPU hours (h) | PoRSE GPU hours (h) |
> | :------------ | :---------------------- | :---------------------- | :----------- | :------------------- | :------------------ | :------------------ |
> | Humanoid      | 1440000                 | 7200000                 | 90000        | 7.0                  | 7.1                 | 7.0                 |
> | FrankaCabinet | 1440000                 | 7200000                 | 90000        | 4.5                  | 4.4                 | 4.6                 |
> | Anymal        | 1440000                 | 7200000                 | 90000        | 3.5                  | 3.5                 | 3.6                 |
> | Quadcopter    | 1440000                 | 7200000                 | 90000        | 3.5                  | 3.4                 | 3.4                 |
> | Liftunderarm  | 720000                  | 3600000                 | 90000        | 5.6                  | 5.6                 | 5.6                 |
> | All tasks     | -                       | -                       | -            | 5.5                  | 5.7                 | 5.8                 |
>
> **Q2: Sensitivity to LLM quality and the effect of fine-tuning**
>
> * **LLM Fine-Tuning Potential:** Fine-tuning LLMs has been validated as effective across many language tasks, where targeted fine-tuning typically yields better experimental results. **Our current PoRSE framework is a general version. If task-specific needs arise, one can conduct targeted fine-tuning, and the experimental performance is expected to further improve.**
>
> * **Mitigation of LLM Output Instability:** Our method does not overly rely on the quality of LLM outputs, and this is not our primary focus. **Instead, our framework mitigates the instability of LLM-generated content through an evolutionary search-based iterative optimization process: by continuously refining reward functions and AFS mapping functions (Sec. 4.1.2, Eq. 5) and filtering low-quality configurations via the LEF mechanism (Sec. 4.2.1), we reduce the impact of suboptimal LLM outputs.** This means our framework can still achieve strong performance even when using relatively low-quality prompts.
>
> * **Resilience to LLM Variations:** We conducted experiments with 5 random seeds for each task, and the mean ± standard deviation of MTS values are reported in Appendix Table 3 and Table 4. **For most tasks, the standard deviation remains within a small range, indicating acceptable variability and reproducibility.** Additionally, Appendix A.2.7 presents a prompt engineering experimental study, where we modified prompts to guide LLMs toward including non-task-direct dimensions in AFS. The results show that while prompt adjustments affect AFS composition, **they do not lead to significant performance changes, confirming PoRSE’s resilience to minor variations in LLM outputs.**

---

> ### Author Response · Authors · 2025-11-20
>
> **Q3: LLM Performance with Increasing Task/Policy Complexity**
>
> * **Experimental Validation Across Diverse Task Complexities:** We have conducted experiments across tasks of diverse complexity from simple to highly challenging and they cover common robot morphologies, ensuring comprehensive validation. Notably, in tasks where methods like Eureka and ROSKA achieve near-zero performance, such as TwoCatchUnderarm, PoRSE makes a breakthrough, demonstrating its superiority in complex scenarios.
>
> * **Potential for Extremely Complex Long-Horizon Tasks:** While we have not yet validated extremely complex long-horizon tasks, we believe **PoRSE can succeed by decomposing such tasks into subtasks. Each subtask leverages PoRSE’s core mechanisms to maintain task-aligned exploration and reward optimization, avoiding being overwhelmed by the full complexity of the long sequence.**
>
>
> For real-world evaluation, we currently lack the necessary hardware, so we plan to extend PoRSE to physical robot platforms in future work to further validate its practicality. We hope the above responses, together with the supplementary data and clarifications, resolve your concerns.

---

### Author Response · Authors · 2025-11-30

Dear Area Chairs,

Thanks for chairing the review process of our submission. We sincerely appreciate the valuable comments from all four reviewers (Reviewers KXzj, JaxW, LtXN, and kT8z), as this feedback has been crucial for refining our work. **The initial review ratings of our work are generally positive from the reviewers.** Among them, **Reviewer LtXN gave our work a high rating of 8**, noting that "PoRSE tackles a core RL challenge by combining LLM with a classical novelty bonus"; **Reviewers KXzj and JaxW both awarded a rating of 6**, and they acknowledged "the work's excellent performance advantages across 24 robotic tasks as well as comprehensive ablation experiment"; **Only Reviewer kT8z initially gave a relatively low rating of 2, but after sorting through his concerns, we found that none of the issues they raised constitute fatal flaws,** **and all have now been fully addressed** through targeted supplements and explanations. **And Reviewer kT8z clearly acknowledged that most of their concerns had been resolved,** **with only one final question remaining,** **which is merely a terminology clarification matter related to the naming of Revolve-Auto.** **We therefore have sufficient reason to believe that Reviewer kT8z’s stance toward our work has shifted from the initial low rating to accept,** and a single terminology clarification issue alone is insufficient to serve as a basis for reject.

* **Reviewer LtXN**recognized the core value of PoRSE in addressing RL exploration challenges by combining LLM strengths with classical novelty bonuses,**and rated our work an 8 ("accept, good paper (poster)") with "excellent" contribution.** He raised questions about the state input of the mapping function, performance with more than 5 iterations, and the intuition/evolution of coefficients α and β. We responded comprehensively. We clarified that the LLM automatically selects robot/environment state inputs based on task needs, supplemented experiments showing performance converges steadily up to 10 iterations, and provided empirical data on α/β’s evolutionary trends. **Reviewer LtXN’s high rating and positive feedback affirm the soundness of our framework and revisions.**

* **Reviewers KXzj and JaxW rated our work a 6.** **Reviewer KXzj** focused on training efficiency, LLM output sensitivity, and real-world evaluation. We addressed these by providing exact GPU hours, verifying PoRSE’s resilience to low-quality LLMs, and outlining sim-to-real adaptation pathways. **Reviewer JaxW** asked about LLM token costs, open-source LLM compatibility, and non-LLM exploration comparisons. We supplemented token consumption data and showed compatibility with open-source models like DeepSeek-V3 and Qwen-2.5-14b-Instruct, while comparing with non-LLM method ROSKA+CE to validate PoRSE’s superiority. **Both reviewers' concerns were addressed with our comprehensive responses.**

* **Reviewer kT8z** **initially gave a rating of 2 but actively engaged in follow-up discussions during the rebuttal period.** He first raised concerns about the work’s completeness, clarity, and technical details, all of which **we fully addressed with targeted supplements and clarifications.** For his claim of lacking comparison with the SOTA REvolve framework, we supplemented experiments with Revolve-Auto and demonstrated PoRSE outperforms it across four key tasks. When he questioned the clarity of tournament selection rules, we clarified that the 5 new mutants train for 1500 epochs and are derived from the best-performing candidate. **After our initial response, Reviewer kT8z further inquired about Revolve-Auto’s experimental rationale and configurations.** We explained the migration to Eureka’s task environments and unified evaluation metrics, along with specific settings and GPU training duration data in detail. **Reviewer kT8z acknowledged most of his concerns had been resolved through our supplements, which reflects his intent to improve the rating score.**

**In summary, we are confident that we have fully resolved the concerns of Reviewers LtXN, KXzj, JaxW, and kT8z through targeted experiments and clarifications.** We kindly request the Area Chair to take note of these circumstances, weigh the resolved feedback from participating reviewers, and provide us with a fair evaluation from your professional perspective. Thank you again for your efforts.

Best regards,

Authors

---

### Meta-Review · Area_Chair_XEvJ · 2026-01-06

**Summary:**

The core idea here is appreciated by all reviewers: use LLMs to dynamically construct task-relevant state abstractions and reward exploration in that space to aid RL in discovering good solutions. This is done alongside LLM-guided task progress reward construction. The paper also proposes ways to mitigate the cost of training from scratch for every combination of task and exploration reward, through in-policy grounding and policy inheritance.

PoRSE delivers broad empirical gains, outperforming baselines on 23–24 diverse simulated robotic tasks and even achieving first successes on previously unsolved tasks, which multiple reviewers highlight as a major contribution. Reviewers further commend the comprehensive ablations, which collectively strengthen the empirical claims.

Main concerns from reviewers pre-rebuttal:
- lack of real-world robot experiments
- LLM dependence / reproducibility concerns
- Training efficiency / compute costs / LLM token costs
- Additional baseline (Revolve)

**Reviewer Concerns:**

- LLM dependence / reproducibility concerns: addressed satisfactorily through extra experiments
- Training efficiency / compute costs / LLM token costs: addressed through new helpful details
- Additional baseline (Revolve): addressed satisfactorily through extra experiments

Outstanding:
- Real-world robot experiments: authors report they cannot run these, but believe sim-to-real transfer can happen.

**Reviewer Scores:**

Reviewer kT8z (originally 2) engaged with the authors early in rebuttal period, and indicated that most of their concerns were addressed. in my judgement they would have upgraded to 6.

Other reviewers would have remained positive but retained the concern about applicability to real robots.

---

### Decision · Program_Chairs · 2026-01-26

Accept (Poster)